# Nrl knockdown by AAV-delivered CRISPR/Cas9 prevents retinal degeneration in mice

Wenhan Yu[1], Suddhasil Mookherjee[1], Vijender Chaitankar[2], Suja Hiriyanna[1], Jung-Woong Kim[2], Matthew Brooks[2], Yasaman Ataeijannati[1], Xun Sun[2], Lijin Dong[3], Tiansen Li[2], Anand Swaroop[2] & Zhijian Wu[1]

In retinitis pigmentosa, loss of cone photoreceptors leads to blindness, and preservation of cone function is a major therapeutic goal. However, cone loss is thought to occur as a secondary event resulting from degeneration of rod photoreceptors. Here we report a genome editing approach in which adeno-associated virus (AAV)-mediated CRISPR/Cas9 delivery to postmitotic photoreceptors is used to target the Nrl gene, encoding for Neural retina-specific leucine zipper protein, a rod fate determinant during photoreceptor development. Following Nrl disruption, rods gain partial features of cones and present with improved survival in the presence of mutations in rod-specific genes, consequently preventing secondary cone degeneration. In three different mouse models of retinal degeneration, the treatment substantially improves rod survival and preserves cone function. Our data suggest that CRISPR/Cas9-mediated NRL disruption in rods may be a promising treatment option for patients with retinitis pigmentosa.

[1] Ocular Gene Therapy Core, National Eye Institute, NIH, 6 Center Drive, Room 307, Bethesda, Maryland 20892, USA. [2] Neurobiology-Neurodegeneration and Repair Laboratory, National Eye Institute, NIH, 6 Center Drive, Room 307, Bethesda, Maryland 20892, USA. [3] Genetic Engineering Core, National Eye Institute, NIH, 6 Center Drive, Room B102, Bethesda, Maryland 20892, USA. Correspondence and requests for materials should be addressed to Z.W. (email: wuzh@mail.nih.gov).

Retinitis pigmentosa (RP) is characterized by progressive retinal degeneration and is a leading cause of inherited blindness, afflicting 1 in 4,000 live births[1]. To date, more than 3,000 mutations in over 60 genes have been causally associated with RP[2] (https://sph.uth.edu/retnet/). Genetic defects in a majority of RP genes initially lead to rod photoreceptor dysfunction or pathology[3]. Rods are dim-light-sensing neurons distributed throughout the retina with the highest densities just outside of the macula[2,4], whereas cones respond to higher levels of light and mediate colour vision. Cone photoreceptors are present at low density throughout the retina and at high density at the centre of macula (called fovea), which is characterized by absence of rods and responsible for the highest visual acuity[5]. A typical RP patient initially manifests night-blindness with gradual constriction of the visual field but sparing of central vision. As the rod loss progresses secondary death of cones ensues, leading to deterioration of visual acuity and eventual blindness. The mechanisms underlying the secondary cone cell death are poorly understood, and no effective therapy is currently available for patients.

One treatment approach for inherited eye disorders consists in correcting the underlying molecular defects using virally mediated gene replacement, as exemplified by recent clinical trials for Leber's congenital amaurosis[6] and choroideremia[7]. This strategy has also been tested for RP in several animal models of recessive disease with varying degree of success[6]. For RP caused by a dominant mutation, inactivation of the mutant allele has been evaluated in disease models using ribozymes[8] and RNA interference[9], and by transcriptional repression using zinc finger-based approaches[10,11]. Though encouraging, these gene-specific approaches appear to be less practical due to extensive heterogeneity in the genetic defects underlying inherited eye disorders. Therefore, gene-independent approaches targeting common disease pathways are being actively pursued[12,13]. As cone photoreceptors support daytime vision and visual acuity, preserving cone function and viability is critical to the quality of life of RP patients. One interesting concept that has been proposed consists in reprogramming adult rod photoreceptors by ablating neural retina leucine zipper (Nrl)[14], a transcription factor that specifies rod cell fate during retinal development and plays a central role in maintaining rod homeostasis in mature retina[3,15,16]. Ablation of Nrl in adult rods leads to loss of rod features and acquisition of cone characteristics. This results in a consequently improved survival in the presence of rod-specific gene mutations, presumably preventing secondary cone loss[14]. However, this proof-of-principle study was conducted in Nrl-floxed mouse lines using inducible Cre-mediated gene knockout (KO), which is not a practical therapeutic approach. Additionally, therapeutic effects were only evaluated in rhodopsin-KO mice. Whether this approach could be generalized to retinal disorders caused by dominant mutations, or mutations in genes other than rhodopsin, requires further evaluation.

Commonly used gene knockdown techniques, including antisense oligonucleotides, ribozymes and RNA interference, target mRNA molecules and as such, cannot completely abolish gene expression. In contrast, the CRISPR (clustered regularly interspaced short palindromic repeats)/Cas9 gene editing system can efficiently disrupt genes at desired loci, enabling complete gene KO[17]. In this system, a single guide RNA (sgRNA) directs the Cas9 endonuclease to specific sites in the genome proximal to a protospacer adjacent motif, causing a double-strand break (DSB). Host cells efficiently repair DNA damage via the non-homologous end joining pathway, and during this process introduce insertions or deletions (indels) at the target site. Gene disruption can thus be achieved if the target site is within the coding sequence and indels lead to frameshifts. CRISPR/Cas9-mediated gene disruption is rapidly becoming a routine method for creating gene knockouts in cell lines and in animal models[17].

A large number of gene mutations in inherited retinal diseases call for the therapeutic use of CRISPR/Cas9 genome editing, as it could potentially provide a stable and long-term therapeutic benefit. Towards this goal, in vivo knockdown of a mutant rhodopsin gene has been conducted in rats[18], with CRISPR/Cas9 components delivered by electroporation at postnatal day 0 (P0) when rods are proliferating. More recently, adeno-associated virus (AAV)-mediated CRISPR/Cas9 was used for targeted gene disruption in retinal ganglion cells following intravitreal vector administration in mice[19]. However, in vivo delivery of CRISPR/Cas9 to postmitotic photoreceptors, which is therapeutically more relevant for a majority of patients with inherited retinal degeneration, has not been reported.

In this study, we establish an AAV-based CRISPR/Cas9 system for targeted gene disruption in postmitotic photoreceptors, and validate this approach by performing in vivo knockdown of Nrl in the retina. Our studies show that loss of Nrl expression in rods causes them to acquire cone-like characteristics, presents with increased survival and leads to preserved cone function in three independent models of retinal degeneration caused by rod-specific gene mutations.

## Results

**An AAV-delivered photoreceptor-specific CRISPR/Cas9 system.** Expression cassettes of SpCas9 and sgRNA were delivered by two separate AAV vectors (Fig. 1a). The Streptococcus pyogenes Cas9 (SpCas9) coding sequence was placed in between a photoreceptor-specific human rhodopsin kinase (RK) promoter[20] and a synthetic polyadenylation signal. A human RNA polymerase III promoter U6 was used to drive sgRNA expression. A tdTomato expression cassette was included in the sgRNA vector to track transduction (Fig. 1a). These vectors were packaged into AAV type 8 (AAV8), a serotype transducing mouse photoreceptors efficiently[21]. Substantial expression of tdTomato and Cas9 in mouse retina was observed following vector delivery (Supplementary Figs 1a,b and 23). The RK promoter-driven tdTomato expression was limited to photoreceptors, confirming the photoreceptor specificity of this AAV-CRISPR/Cas9 system (Supplementary Fig. 1c). Persistent expression of Cas9 nuclease did not seem to affect retinal function even at the highest vector dose ($5 \times 10^9$ vector genomes) (Supplementary Fig. 2).

Knockdown efficiency of the AAV-CRISPR/Cas9 system was evaluated using enhanced green fluorescent protein (EGFP) gene as a target. We first searched for protospacer adjacent motif sequences within EGFP coding region[22] and identified 119 potential 20-nucleotide protospacer sequences. Five sequences with leading scores and three that were previously reported[23,24] (ET1 to ET8, see Supplementary Table 1 and Supplementary Fig. 3a) were assembled into the sgRNA constructs (Fig. 1a). Two of the sgRNAs (containing ET1 and ET3, respectively) exhibited higher ability to form indels than the rest (Supplementary Fig. 3b) and were able to abolish EGFP expression in cell culture efficiently (Supplementary Fig. 3c–f). One of them (containing ET3, Fig. 1b) was packaged into AAV8 (designated AAV-sgRNA-EGFP) and was coinjected with AAV-Cas9 into subretinal space of 2-week-old Nrl-L-EGFP mice, which specifically express EGFP in rods (Fig. 1c)[25]. At P14, mouse photoreceptors are postmitotic though not completely mature. At 2.5 months post injection, efficient indel formation ($11.95 \pm 0.1\%$) was detected in fluorescence-activated cell sorting (FACS)-enriched tdTomato-positive photoreceptors by SURVEYOR assay (Fig. 1d). In control

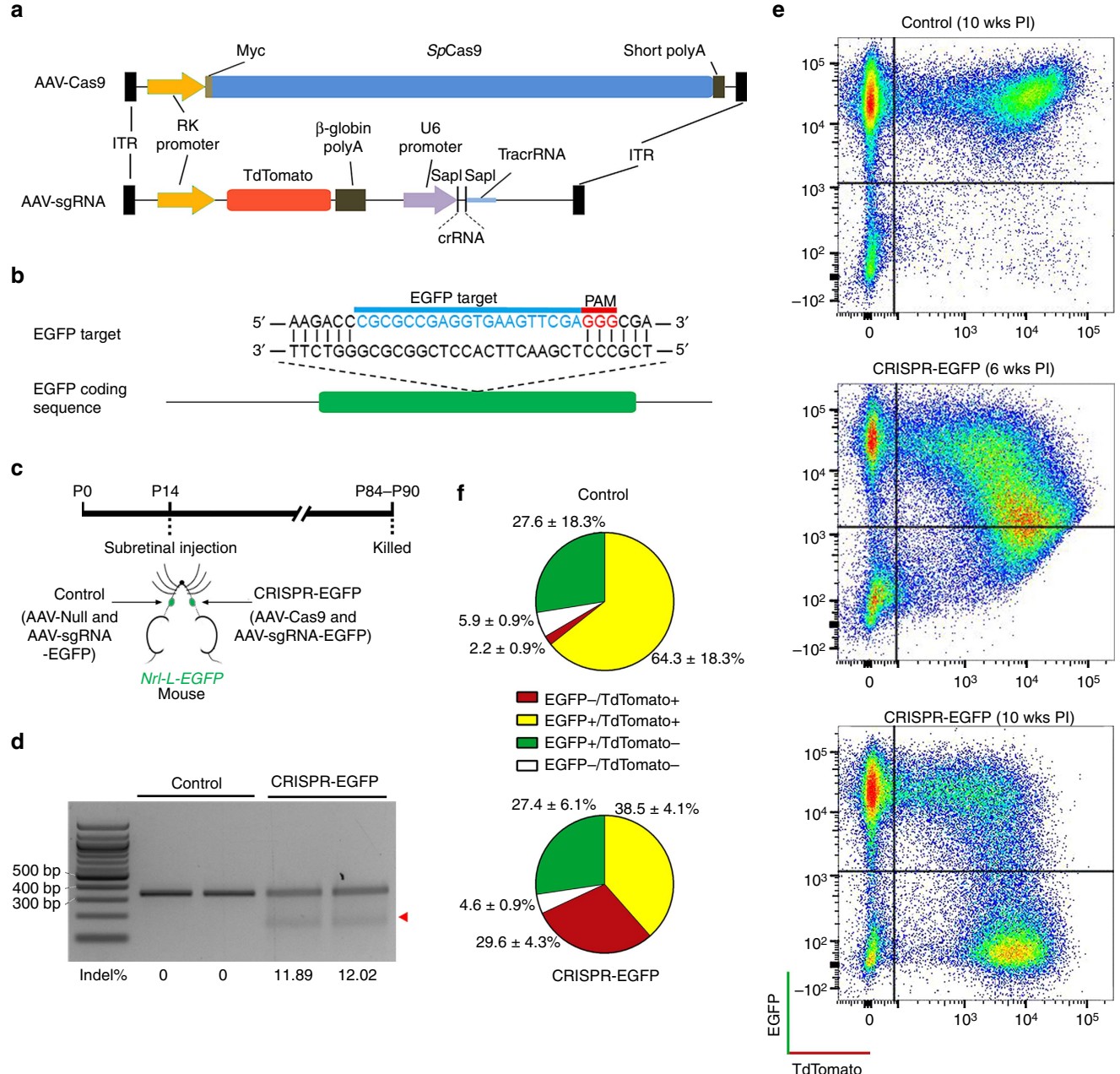

**Figure 1 | EGFP knockdown in photoreceptors by an AAV-CRISPR/Cas9 system. (a)** Schematic representation of the AAV vectors delivering SpCas9 and sgRNA. (**b**) Schematic representation of the EGFP locus showing the location of the sgRNA target. The targeted genomic site is indicated in blue. Protospacer adjacent motif (PAM) sequence is marked in red. (**c**) Timeline for the EGFP knockdown experiments. Each *Nrl-L-EGFP* mouse was subretinally injected with $2.5 \times 10^9$ vector genomes (vg) AAV-Cas9 and $2.5 \times 10^9$ vg AAV-sgRNA-EGFP in one eye, and the same doses of AAV-Null and AAV-sgRNA-EGFP (control) in the fellow eye. A total of eight mice including both genders were used. (**d**) SURVEYOR nuclease assay revealing indel formation at the EGFP locus in flow-sorted tdTomato-expressing cells. Each lane contained sample from an individual mouse ($n = 2$). DNA fragments digested by SURVEYOR nuclease are indicated by arrowhead. Indel rate of each sample is shown below the gel image. (**e**) Representative FACS plots of dissociated cells from retinas receiving control vectors (upper panel) or CRISPR-EGFP vectors (middle and lower panels). Dissociated cells from two retinas were used in each group. (**f**) Statistical analysis of flow-sorted retinal cells of three independent experiments. Data are represented as mean ± s.d. PI, post injection.

retinas receiving co-delivery of AAV-sgRNA-EGFP and an AAV-Null vector, EGFP-expressing rods accounted for 91.9 ± 0.1% of the total dissociated cells, while tdTomato-positive cells indicating sgRNA transduction accounted for 66.6 ± 27.2% (Fig. 1e,f). The sgRNA-transduced cells were predominantly rods, as 96.8 ± 0.6% of the tdTomato-positive cells were also EGFP positive. In the retinas receiving codelivery of AAV-sgRNA-EGFP and AAV-Cas9, the percentage of EGFP-expressing cells decreased

to 65.9 ± 5.1%, while the transduction rate remained similar (68.1 ± 5.9% tdTomato-positive cells), indicating that ∼30% of total rods abolished EGFP expression. Therefore, successful ablation of the EGFP gene happened in estimated 43% of the sgRNA-transduced rods. Lack of EGFP ablation in the rest 57% of the sgRNA-transduced cells could be caused by lack of Cas9 expression, in-frame indels unable to abolish EGFP expression and/or multiple copies of EGFP transgene in the mice[25]

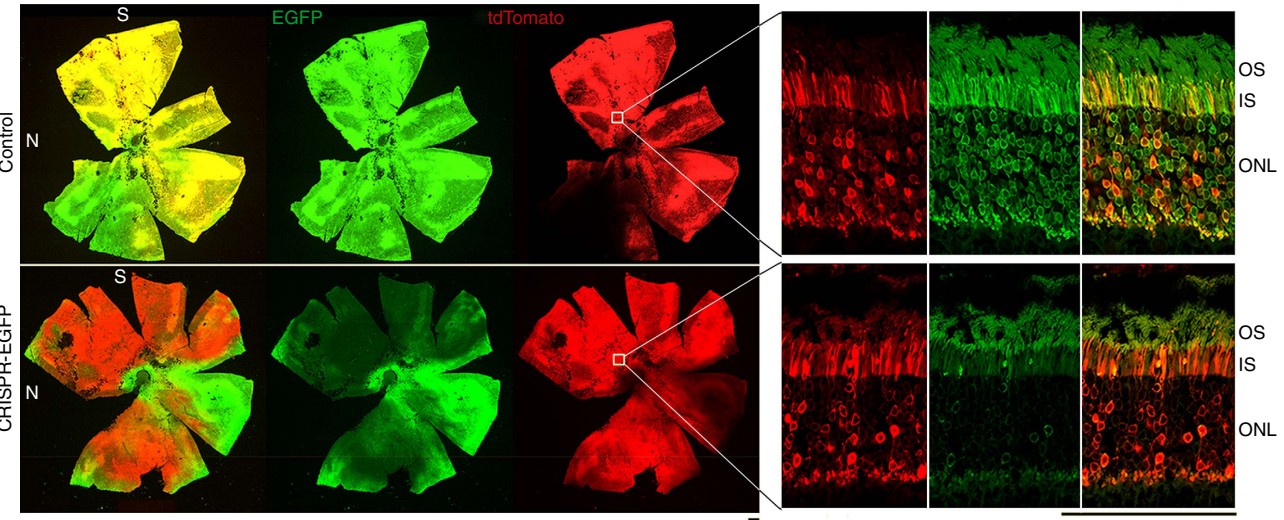

**Figure 2 | EGFP knockdown by AAV-CRISPR/Cas9 in mouse retina.** Four *Nrl-L-EGFP* mice including both genders received subretinal codelivery of $2.5 \times 10^9$ vector genomes (vg) AAV-Cas9 and $2.5 \times 10^9$ vg AAV-sgRNA-EGFP per eye at P14. Same doses of AAV-Null and AAV-sgRNA-EGFP were coinjected to the fellow eyes as controls. Representative retinal whole-mounts (left panels) and sections (right panels) of a treated mouse at 3 months of age are shown. TdTomato expression (red) indicates the transduction by the AAV-sgRNA vector. IS, inner segments; N, nasal; ONL, outer nuclear layer; OS, outer segments; S, superior. Scale bars, 50 μm.

exceeding the capacity of CRISPR-mediated gene disruption. The FACS gating strategy is shown in Supplementary Fig. 4. Retinal whole-mounts and sections further validated EGFP knockdown in rods (Fig. 2).

***Nrl* knockdown in postmitotic mouse photoreceptors**. Among the five designed sgRNAs (containing protospacer sequences NT1 to NT5, respectively) against the mouse *Nrl* coding region (Supplementary Fig. 5a), the NT2-containing sgRNA (Fig. 3a) was chosen for in *vivo* study based on its relatively higher ability to generate indels (Supplementary Fig. 5b), and lower predicted off-target potential. The vector was packaged into AAV8 and designated AAV-sgRNA-*Nrl*.

A combination of AAV-sgRNA-*Nrl* and AAV-Cas9 (designated CRISPR-*Nrl*) was injected subretinally into wild-type (WT) C57bl/6j mice at P14 (Fig. 3b). The fellow eyes were injected with a combination of AAV-sgRNA-EGFP and AAV-Cas9 (designated CRISPR-EGFP) as controls. Cotransduction of AAV vectors was observed in a majority of photoreceptors following subretinal delivery of two reporter AAV vectors (Supplementary Fig. 6). At 11 to 13 weeks post injection, the SURVEYOR assay revealed formation of indels ($4.1 \pm 0.6\%$) in FACS-enriched tdTomato-positive cells (Fig. 3c). Deep sequencing of the sgRNA-*Nrl* region indicated that 98% of total reads included changes almost exclusively at the targeted genome site (Fig. 3d). The sequence alterations were generally limited to one or two nucleotides indels, resulting in frameshifts. Interestingly, one adenosine insertion accounted for over 80% of the total reads (Fig. 3e). This could explain the relatively low indel rate detected in the SURVEYOR assay (Fig. 3c), as Surveyor nuclease only cleaves mismatched heteroduplexes after denaturing and annealing of the target amplicons, which were the minority species in this case. To determine potential off-target events, we performed deep sequencing at 10 predicted off-target sites, 4 of which are top ranking and 6 are exon located. None of these sites revealed significantly higher rate of sequence alterations compared with the background in untreated or CRISPR-EGFP-treated eyes (Supplementary Table 2). As AAV-delivered Cas9 and sgRNA-*Nrl* were likely persistently expressed, long-term on-target and

off-target events were examined by deep sequencing at 9.5 months post treatment (Supplementary Fig. 7). The results revealed a slightly lower on-target mutation rate ($\sim 93\%$) and a similar pattern of sequence alterations with a predominant one adenosine insertion ($\sim 88\%$), compared with those at 3 months post treatment. Mutation rates at the predicted potential off-target sites were again not significantly higher than those in control groups (Supplementary Table 2), suggesting that long-term expression of CRISPR components may not necessarily impose a higher risk of off-targeting if the sgRNA is appropriately selected.

Substantial reduction of the NRL protein was observed in the CRISPR-*Nrl*-treated retina (Fig. 3f,g, Supplementary Figs 8 and 21). In control retina, nuclei in the outer nuclear layer (ONL) were largely positively stained for both NRL and CRX. As NRL is rod-specific whereas CRX is expressed in both rods and cones, a few CRX-positive NRL-negative cell bodies in the outer part of ONL were likely cone nuclei. In contrast, a majority of ONL cell bodies in CRISPR-*Nrl*-treated retina were not stained for NRL but were positive for CRX, indicating successful *Nrl* ablation. The deep sequencing (Fig. 3d,e) and the immunofluorescence analyses (Fig. 3g) collectively suggest that two *Nrl* alleles were disrupted in a majority of CRISPR-*Nrl*-transduced cells.

**Phenotype changes following *Nrl* knockdown**. Although outer segments (OS), inner segments (IS) and ONL appeared normal, the retinas receiving CRISPR-*Nrl* broadly exhibited larger nuclei compared with the control retinas (Fig. 4a). In particular, some photoreceptor nuclei had smaller heterochromatin regions unlike typical rod nuclei and similar to that of mouse cones. Thus, changes in the chromatin architecture in response to *Nrl* ablation in some postmitotic rods suggest a movement towards cone-like cells, consistent with S-cone being the default pathway[3]. To validate these changes, we conducted *Nrl* knockdown in *Crxp-Nrl* mice having a rod-only retina[26] and performed ultrastructural analysis by electron microscopy (Fig. 4b). In contrast to rods in the control retina with small nuclei and a large mass of heterochromatin, the CRISPR-*Nrl*-treated retinas revealed some photoreceptors with cone-like morphology including bigger

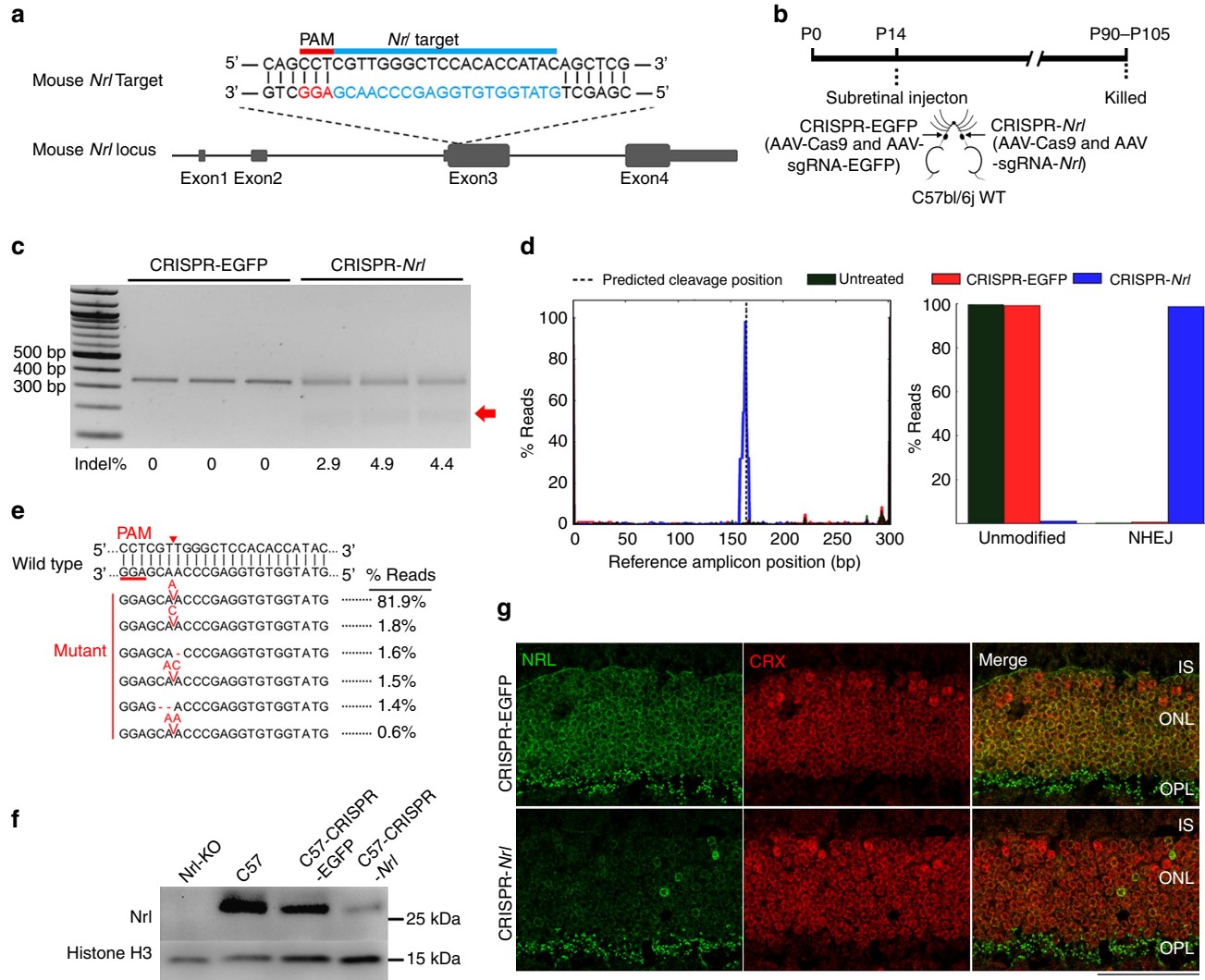

**Figure 3 | *Nrl* gene knockdown in postmitotic photoreceptors using AAV-CRISPR/Cas9.** (**a**) Schematic representation of mouse *Nrl* locus showing the location of the sgRNA target. The targeted genomic site is indicated in blue. Protospacer adjacent motif (PAM) sequence is marked in red. (**b**) Timeline for the *Nrl* knockdown experiments. Each wild-type (WT) C57bl/6j mouse was subretinally injected with $5 \times 10^9$ vector genomes (vg) AAV-Cas9 and $2.5 \times 10^9$ vg AAV-sgRNA-*Nrl* (CRISPR-*Nrl*) in one eye and same doses of AAV-Cas9 and AAV-sgRNA-EGFP (CRISPR-EGFP) in the fellow eye. A total of eight mice including both genders were used. (**c**) SURVEYOR nuclease assay showing indel formation in the *Nrl* locus in flow-sorted tdTomato-expressing cells. Each lane contained sample from an individual mouse ($n = 3$). DNA fragments digested by SURVEYOR nuclease are indicated by red arrow. Indel rate of each sample is shown below the gel image. (**d**) Rate of sequence change at the target site of *Nrl* locus in flow-sorted tdTomato-expressing cells by deep sequencing analysis. Schematic positions of indels in the amplicon (left) and total non-homologous end joining (NHEJ) frequencies (right) of untreated (green), CRISPR-EGFP-treated (red) and CRISPR-*Nrl*-treated (blue) eyes were compared. Two retinas were used for each treatment. (**e**) Representative mutation patterns and corresponding ratios in total reads detected by deep sequencing of *Nrl* locus. Top, wild-type sequence; red dashes, deleted bases; red bases: insertions; the red triangle indicates CRISPR/Cas9 cutting site. (**f**) Immunoblot analysis of NRL protein at 12 weeks post vector injection. Retinas from *Nrl*-knockout (KO) and C57bl/6j (C57) mice served as negative and positive controls, respectively. Histone H3 served as loading control. (**g**) Immunostaining of NRL in retina sections. IS, inner segments; ONL, outer nuclear layer; OPL, outer plexiform layer; Scale bar, 50 μm.

nuclei with much larger euchromatin domain[27], consistent with the observations by light microscopy.

No obvious differences were observed between CRISPR-*Nrl*-treated and control retinas in bipolar cells and post-photoreceptor synapses (Supplementary Fig. 9a,b), similar to previous findings in *Nrl* − / − mice[28]. Location of retinal ganglion cells (Supplementary Fig. 9c), integrity of retinal pigment epithelium (RPE) (Supplementary Fig. 10a,b) and retinal vasculature (Supplementary Fig. 11) were not noticeably altered by CRISPR-*Nrl* treatment, in contrast to those observed in *Nrl* − / − mice[29]. Mild gliosis was observed in both CRISPR-

EGFP- and CRISPR-*Nrl*-treated retinas (Supplementary Fig. 10c), indicating stress response likely caused by Cas9 expression and/or by DSB creation and repair. However, location of Muller cell nuclei was not altered (Supplementary Fig. 10c), different from that of *Nrl* − / − retina[29]. These results collectively indicate that CRISPR-*Nrl* treatment does not significantly alter the overall retinal structure.

Retinal functional changes in treated C57bl/6j mice were monitored by electroretinography (ERG) (Fig. 4c,d). Markedly reduced amplitudes of dark-adapted a-wave and b-wave were observed in the eyes receiving CRISPR-*Nrl*, indicating

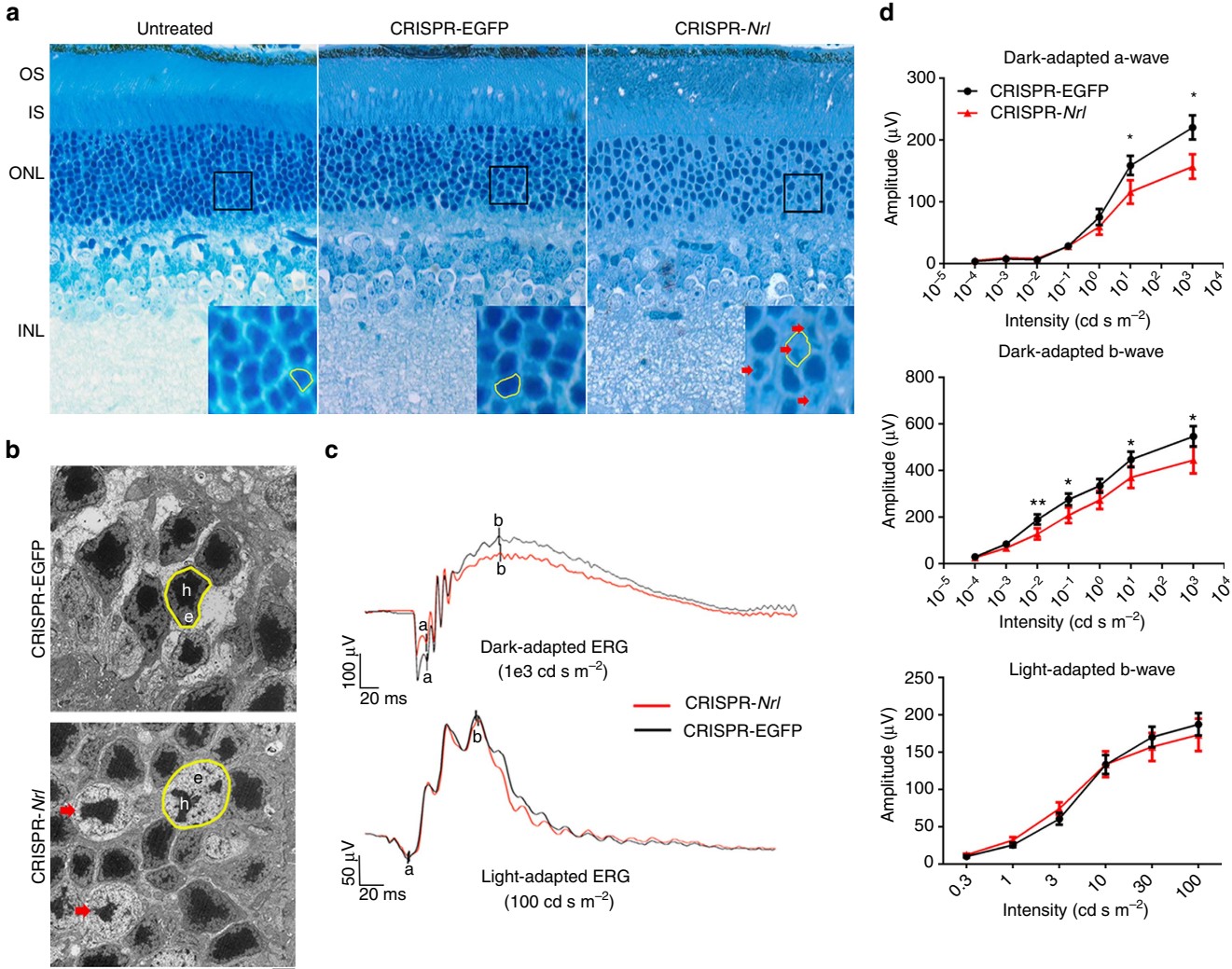

**Figure 4 | Morphological and functional changes of retina following _Nrl_ knockdown. (a)** Toluidine blue-stained semi-thin retinal sections (0.5 μm) of untreated, CRISPR-EGFP-treated and CRISPR-_Nrl_-treated eyes of C57bl/6j mice. Insets show the magnified images of the marked areas. **(b)** Electron micrographs of photoreceptor nuclei in CRISPR-EGFP- and CRISPR-_Nrl_-treated _Crx_p-Nrl (rod-only) mice. In **a,b**, sample nuclei are outlined in yellow. Heterochromatin (h) and euchromatin (e) are labeled. Red arrows indicate cone-like nuclei. Scale bar, 2 μm. **(c)** Representative ERG forms of CRISPR-EGFP- and CRISPR-_Nrl_-treated eyes from a single C57bl/6j mouse at 6 weeks post vector injection. **(d)** Statistical analysis of ERG amplitudes. Significantly lower amplitudes of dark-adapted a- and b-waves were obtained in response to increasing intensities of flash stimuli in the CRISPR-_Nrl_-treated eyes, whereas the amplitude of light-adapted b-wave was not affected ($n = 12$, including both genders). Error bars show s.e.m. and the significance between the CRISPR-EGFP- and CRISPR-_Nrl_-treated eyes was calculated using two-tailed paired _t_-test. *$P < 0.05$; **$P < 0.01$.

compromised rod function. However, no obvious change was detected in the light-adapted b-wave amplitude, suggesting that cone function was relatively stable.

**Alteration of gene expression in _Nrl_-ablated photoreceptors.** We performed global transcriptome analysis (RNA sequencing) for FACS-enriched tdTomato-expressing retinal cells from 2.5-month-old AAV-CRISPR-treated C57bl/6j mice, and compared the result with the transcriptome of mature (P28) WT rods and S cone-like cells from _Nrl_ − / − mouse[30]. Although the tdTomato-expressing cells contained both rods and cones, the transcriptome alteration caused by _Nrl_ ablation would largely represent that happened in rods, as rods outnumber cones by a ratio of 30:1 and appear to be transduced at a similar efficiency as cones by an AAV8 vector[21]. A total of 146 genes exhibited differential expression (90 genes upregulated and 56 genes downregulated) between control (CRISPR-EGFP treated) and

CRISPR-_Nrl_-treated cells (Supplementary Data 1, Supplementary Fig. 12a). In contrast, 6,412 genes were differentially expressed (DE) between mature rods and S cone-like cells[30]. The low number of DE genes following CRISPR-_Nrl_ treatment suggests the lack of plasticity of postmitotic rods for reprogramming into cones. Gene ontology (GO) enrichment analysis of these 146 genes using GOrilla web tool[31,32] revealed 20 enriched GO terms (Supplementary Fig. 12b). Significance of the involvement of these processes following _Nrl_ ablation requires further investigation. One hundred genes were common between DE gene sets of CRISPR-_Nrl_-treated postmitotic and germline _Nrl_ KO photoreceptors and 88 of these exhibited similar up or down expression trends (Supplementary Fig. 13).

We then focused our analysis on the expression changes of photoreceptor-specific genes (Fig. 5a–c). NRL maintains rod phenotype by facilitating rod gene expression synergistically working with CRX and other transcription factors while concurrently repressing cone gene expression through its target

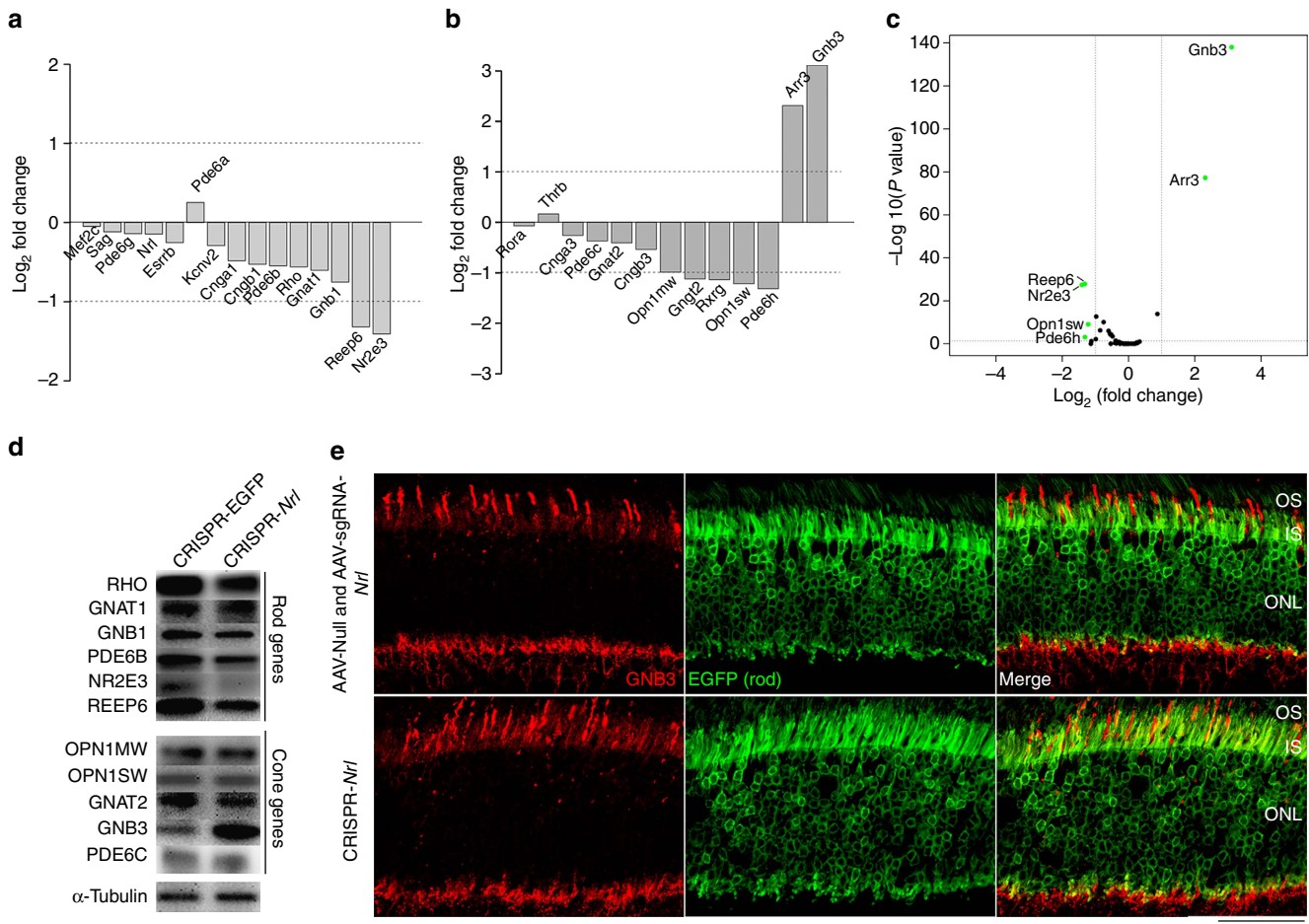

**Figure 5 | Expression alterations following CRISPR/Cas9-mediated *Nrl* knockdown.** (**a–c**) RNA sequencing analysis of flow-sorted tdTomato-expressing cells isolated from C57bl/6j mice receiving subretinal administration of CRISPR-*Nrl* or CRISPR-EGFP (control) vectors, conducted at 2.5 months post vector injection. Four mice including both genders received vector administration. Two retinas were used in each group and the data were averaged from two independent experiments. Fold differences in expression of (**a**) rod- and (**b**) cone-specific genes between CRISPR-*Nrl*- and CRISPR-EGFP-treated eyes are shown. Statistical significance is shown by volcano plot (**c**). Gene expression with fold difference of >2 (absolute log fold difference of >1, indicated by dash lines in **a,b**) and *P* value <0.05 (indicated by dash line in **c**) is considered to be significantly changed, and is shown as green dot. (**d**) Immunoblot analysis of C57bl/6j mouse retinas receiving CRISPR vectors, performed at 2.5 month post vector injection. Combined lysate from two retinas for each treatment was used. (**e**) Immunostaining of GNB3 in retina sections of *Nrl-L-GFP* mice receiving CRISPR-*Nrl* vectors. Control sections were from mice injected with AAV-Null and AAV-sgRNA-*Nrl* vectors. EGFP expressing cells indicate rods. Two mice including both genders were used. Scale bar, 50 μm.

NR2E3 (ref. 3). As predicted, most rod genes were downregulated after *Nrl* knockdown, although a majority showed less than twofold reduction. Two of these, *Nr2e3* and *Reep6*, displayed the strongest reduction, 2.7-fold and 2.5-fold, respectively (Fig. 5a), consistent with previous studies[33,34]. Interestingly, most cone genes were not upregulated (Fig. 5b), probably because of transdifferentiation barriers including epigenetic modifications[14]. However, we observed striking upregulation (8.6-fold and 5.0-fold, respectively) of two cone genes, *Gnb3* and *Arr3*, in *Nrl*-ablated rods, suggesting their specific derepression in the absence of NRL. No significant changes were observed in *Nxnl1* (also known as *RdCVF*) or *Nxnl2* (also known as *RdCVF2*) (Supplementary Fig. 14), genes encoding two rod-derived cone viability factors, which play key roles in cone protection by stimulating glucose metabolism[35,36]. *Grk1*, the gene expressed in both rods and cones, did not show obvious change either (Supplementary Fig. 14).

Significantly lower levels of NR2E3, REEP6 and higher level of GNB3 were detected by immunoblot analysis in retinal lysates after *Nrl* knockdown, with smaller changes observed in other rod-

and cone-specific proteins (Fig. 5d and Supplementary Fig. 22). Immunofluorescence studies demonstrated somewhat lower expression of rhodopsin, PDE6β and REEP6, and minimal changes in S-opsin, cone arrestin and cone phosphodiesterase (PDE) in the CRISPR-*Nrl*-treated retina (Supplementary Fig. 15). The discrepancy between RNA sequencing and immunofluorescence for cone arrestin expression may indicate low sensitivity of antibody. We also detected low-level expression of cone-specific GNB3 in rods following *Nrl* knockdown, but the protein was mislocalized to rod IS, unlike its native OS localization in cones (Fig. 5e).

**Rescue of retinal degeneration following *Nrl* knockdown.** We then tested whether CRISPR-*Nrl* treatment could modify the course of retinal degeneration caused by mutations in rod-specific genes using three distinct mouse models. An initial phase of rod degeneration followed by secondary cone death has been well documented in these mouse lines modelling human RP[37–39] (Figs 6a, 7a and 8a). The mice were subretinally administered

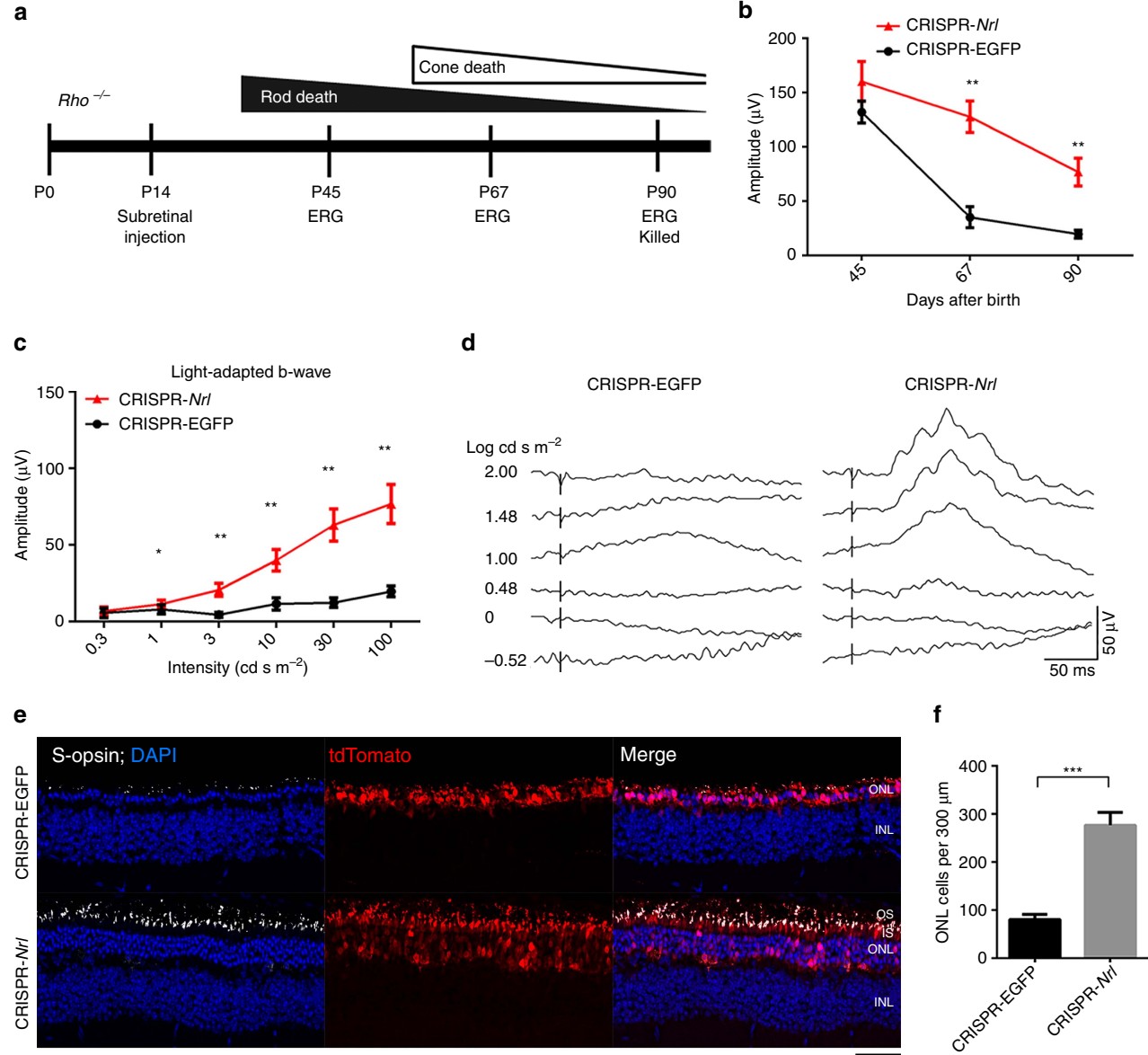

**Figure 6 | Rescue of retinal degeneration in $Rho^{-/-}$ mice following Nrl knockdown.** (**a**) Time course of photoreceptor degeneration in $Rho^{-/-}$ mouse (upper) and timeline for CRISPR/Cas9-mediated Nrl knockdown experiments (lower). Seven mice including both genders received CRISPR-Nrl vector treatment in the right eyes and the control CRISPR-EGFP vector treatment in the left eyes. (**b**) Time-course changes of light-adapted b-wave amplitude in response to the highest stimulus intensity (100 cd s m$^{-2}$) ($n = 7$). (**c**) Light-adapted b-wave amplitude in response to a series of flash stimuli at P90 ($n = 7$). (**d**) Representative ERG waveforms from a single mouse. (**e**) Immunofluorescence images showing better preserved ONL and S-opsin expression in CRISPR-Nrl-treated eyes than CRISPR-EGFP-treated control eyes. Scale bar, 50 μm. (**f**) Quantification of ONL cells in 300 μm segments of retina ($n = 3$). Error bars show s.e.m., and the significance between the CRISPR-EGFP- and CRISPR-Nrl-treated eyes was calculated using two-tailed paired t-test. *$P < 0.05$; **$P < 0.01$; ***$P < 0.001$.

with CRISPR-Nrl vectors at P14, a time point before the onset of rod degeneration. Same doses of CRISPR-EGFP vectors were injected into the control eyes. Photopic ERG responses representing cone function were monitored for 3 months, when rod death is roughly complete in these mouse lines.

Rhodopsin KO ($Rho^{-/-}$) mice were previously used for inducible Cre-mediated Nrl KO[14]. As predicted, the eyes receiving CRISPR-Nrl exhibited much slower decline in b-wave amplitude (Fig. 6b), compared with the control eyes revealing a sharp decline between P45 and P90. The CRISPR-Nrl-treated eyes also displayed better responses under stimuli with a wide range of flash intensities (Fig. 6c,d). In two additional control groups, eyes

receiving either the Cas9 vector or the sgRNA vector did not show ERG rescue (Supplementary Fig. 16a,b). Though rod function was completely lost in mutant retina (Supplementary Fig. 16c,d), rod cell bodies were relatively well preserved following Nrl ablation (Fig. 6e,f and Supplementary Fig. 17a). Cones were protected in the CRISPR-Nrl-treated retina, as shown by substantially more cells with S-opsin and cone PDE localized to the OS (Fig. 6e and Supplementary Fig. 17a). To examine whether treatment given at a later stage could still provide therapeutic benefit, the vectors were injected at P28, a time point when rod degeneration has already started. The results revealed significantly thicker ONL layer and better cone ERG response in

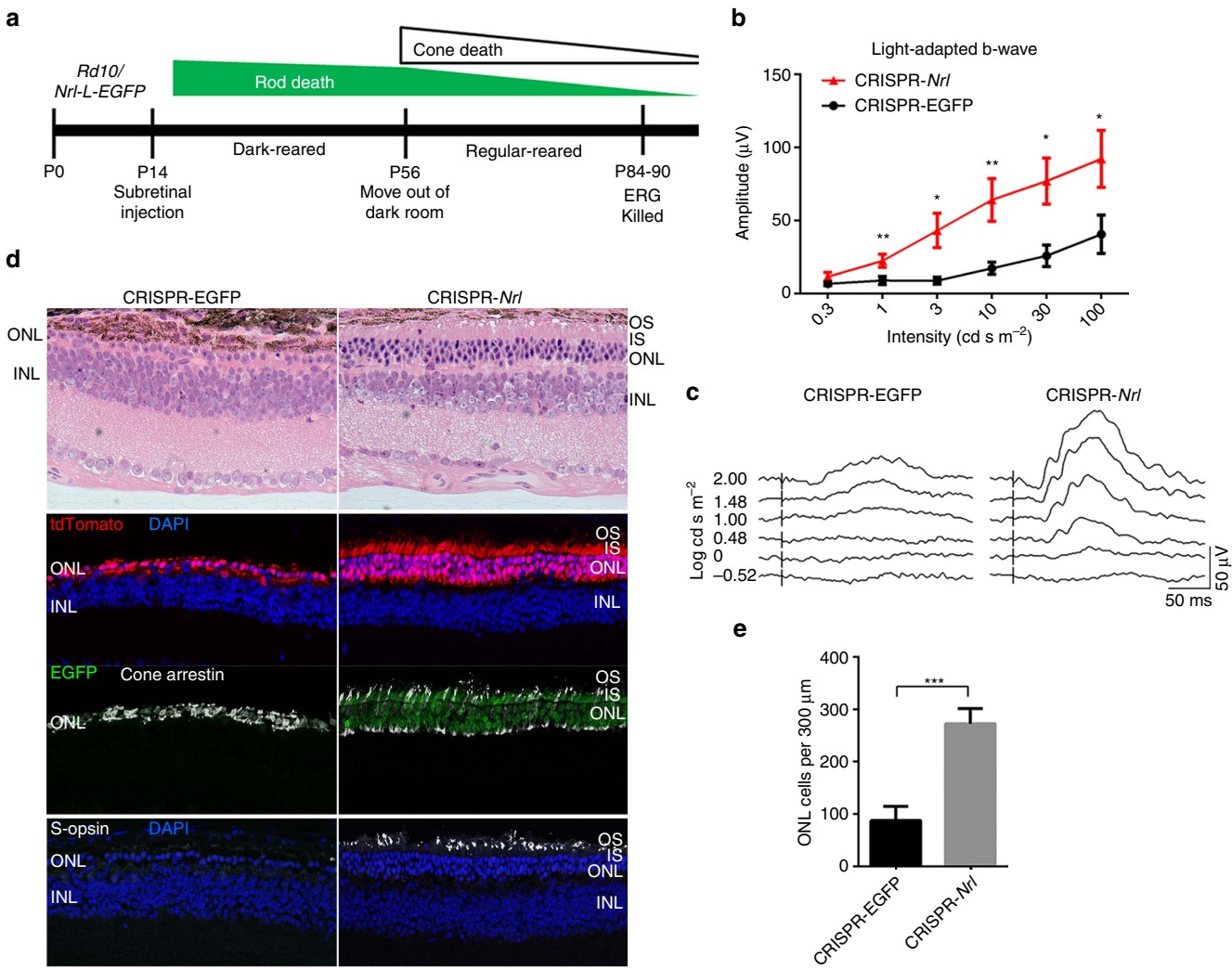

**Figure 7 | Rescue of retinal degeneration in *Rd10/Nrl-L-EGFP* mice following *Nrl* knockdown.** (**a**) Time course of photoreceptor degeneration in *Rd10/Nrl-L-EGFP* (upper) and timeline for CRISPR/Cas9-mediated *Nrl* knockdown experiments (lower). Seven mice including both genders received CRISPR-*Nrl* vector treatment in the right eyes and the control CRISPR-EGFP vector treatment in the left eyes. (**b**) Light-adapted b-wave amplitude in response to a series of flash stimuli at P84 to P90 ($n = 7$). (**c**) Representative ERG waveforms from a single mouse. (**d**) Haematoxylin and eosin (H&E) staining and immunofluorescence images revealing better preserved ONL structure, cone morphology and S-opsin expression in CRISPR-*Nrl*-treated eyes than CRISPR-EGFP-treated control eyes. Scale bar, 50 µm. (**e**) Quantification of ONL cells in 300 µm segments of retina ($n = 3$). Error bars show s.e.m. and the significance between CRISPR-EGFP- and CRISPR-*Nrl*-treated eyes was calculated using two-tailed paired *t*-test. *$P < 0.05$; **$P < 0.01$; ***$P < 0.001$.

the CRISPR-*Nrl*-treated eyes compared with the control eyes (Supplementary Fig. 18), although to a lesser extent than early intervention (Fig. 6 and Supplementary Fig. 17a).

*Rd10* is a naturally occurring mouse line with a hypomorphic mutation in rod-specific *Pde6β* gene[38]. To facilitate tracking of rod photoreceptors, we used *Rd10* line that had been crossed with the *Nrl-L-EGFP* line showing EGFP labelling in all rods. As *Rd10* mice lose rods rapidly, they were dark-reared for 6 weeks after treatment to slow down the degeneration process (Fig. 7a). Similar to the observations in $Rho^{-/-}$ mice, cone function was preserved after *Nrl* ablation as revealed by higher photopic ERG b-wave amplitude (Fig. 7b,c). The rod cell bodies in the vector-transduced area were relatively well maintained even at P90 (Fig. 7d,e and Supplementary Fig. 17b), suggesting that the *Nrl*-ablated rods were able to resist the effect of PDE6β deficiency. Cone viability was revealed by the nearly normal patterns of S-opsin and cone arrestin staining in the CRISPR-*Nrl*-treated retina, in striking contrast to almost complete loss and/or compromised morphology of cones in the control retina (Fig. 7d).

Cone-specific protein GNB3 was again observed in the IS of some EGFP-expressing rods in addition to the OS of remaining cones in the CRISPR-*Nrl*-treated retina (Supplementary Fig. 17b).

*RHO-P347S* mice carrying a mutant human rhodopsin transgene on a WT background[39] were used to examine whether *Nrl* ablation may treat dominant forms of RP. In these mice, rod degeneration is caused by defective vectorial transport of post-Golgi rhodopsin vesicles and is complete within 4 months. The CRISPR-*Nrl*-treated eyes displayed much slower photopic b-wave reduction and significantly better optomotor response than the control eyes (Fig. 8b–e). Histologically, markedly increased number of photoreceptors survived and more cone opsin was preserved in the CRISPR-*Nrl*-treated retinas (Fig. 8f,g and Supplementary Fig. 17c). The treatment appeared able to inhibit the caspase-3-mediated apoptosis pathway of the degenerating photoreceptors, although the caspase-3-independent pathways could also be involved in photoreceptor death in the *RHO-P347S* retina (Supplementary Fig. 19). Collectively, these results showed that rods receiving *Nrl* ablation treatment

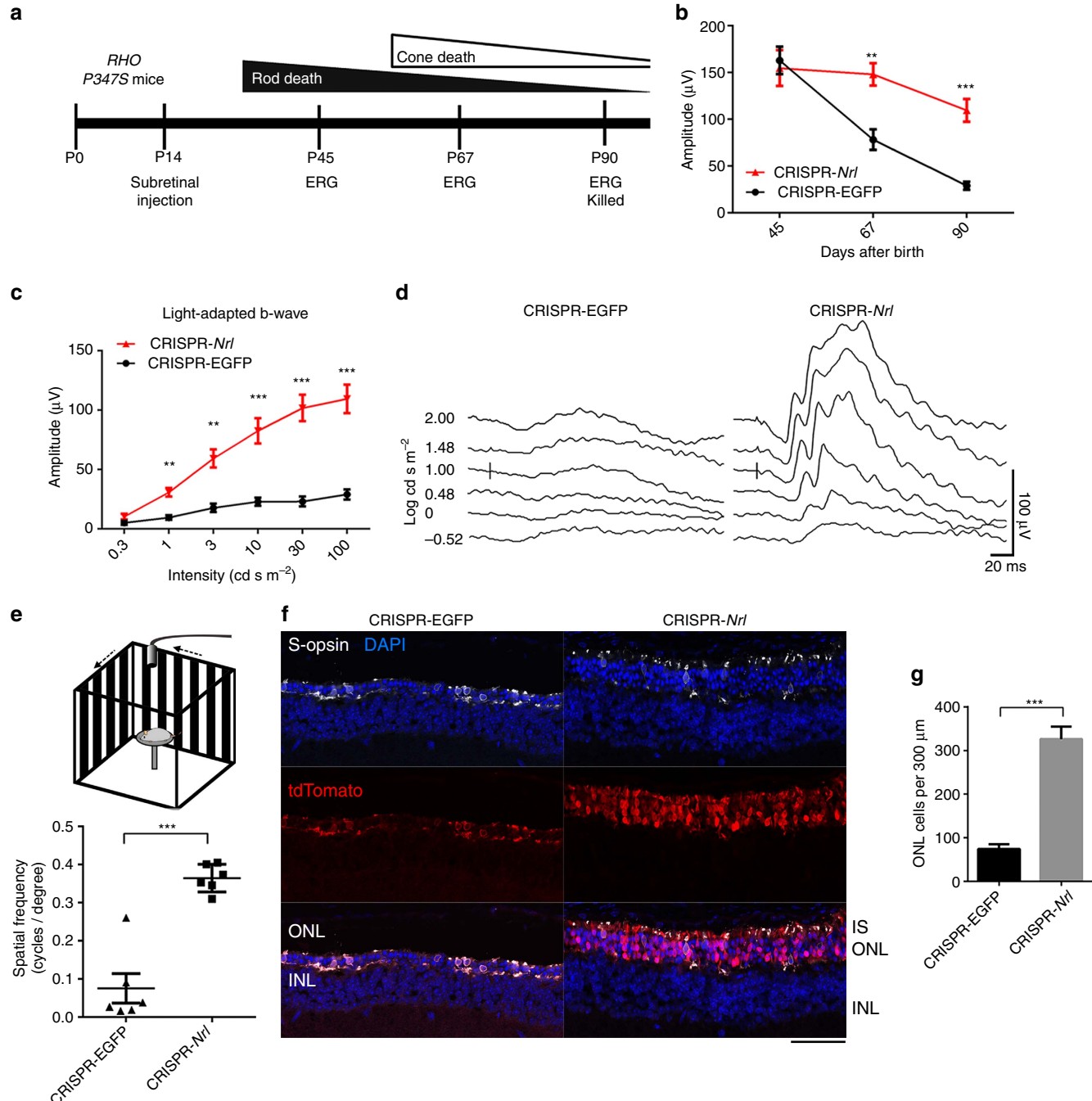

**Figure 8 | Rescue of retinal degeneration in human *RHO P347S* mice by *Nrl* knockdown.** (**a**) Time course of photoreceptor degeneration in human *RHO P347S* transgenic mouse (upper) and timeline for CRISPR/Cas9-mediated *Nrl* knockdown experiments (lower). Ten mice including both genders received CRISPR-*Nrl* vector treatment in the right eyes and the control CRISPR-EGFP vector treatment in the left eyes. (**b**) Time-course changes of light-adapted b-wave amplitude in response to the highest stimulus intensity ($100 \, cd \, s \, m^{-2}$) ($n = 10$). (**c**) Light-adapted b-wave amplitude in response to a series of flash stimuli at P90 ($n = 10$). (**d**) Representative ERG waveforms from a single mouse. (**e**) Schematic of optomotor test (upper) and the spatial resolution (expressed as cycles/degree, lower) from CRISPR-EGFP- and CRISPR-*Nrl*-treated eyes at P90 ($n = 6$). (**f**) Immunofluorescence images revealing better preserved ONL and S-opsin expression in CRISPR-*Nrl*-treated eyes than CRISPR-EGFP-treated control eyes. Scale bar, 50 μm. (**g**) Quantification of ONL cells in 300 μm segments of retina ($n = 3$). Error bars show s.e.m. and paired two-tailed *t*-test was performed for statistical analyses. **$P < 0.01$; ***$P < 0.001$.

were more resistant to deleterious effects of the mutant rhodopsin protein and consequently resulting in prolonged cone survival. In an independent experiment using the three disease models, the delayed cone function loss and prolonged rod survival were observed in each individual mouse at 4 months of age (Supplementary Fig. 20), suggesting that the therapeutic benefit of this approach is long-lasting.

## Discussion

Therapeutic genome editing has long been considered an ideal strategy for permanent correction of genetic defects and is advancing rapidly since the advent of CRISPR/Cas9 technology[40]. Therapeutic application of CRISPR/Cas9 has shown promising outcomes in animal models of several devastating human diseases[41–44]. Inherited retinal diseases could be ideal targets for

*in vivo* CRISPR/Cas9 application since the retina is easily accessible surgically and is isolated by blood–retinal barrier. Additionally, genome modifications can be targeted to specific cell types, and a low amount of vectors carrying CRISPR/Cas9 should be sufficient for disease correction. Precise genome editing following CRISPR/Cas9 treatment is conceptually achievable but occurs at a rate not high enough for therapeutic benefit as of yet since it relies on homology-directed repair, which is unfavourable in postmitotic cells. In contrast, gene disruption or deletion follows the non-homologous end joining pathway with efficiency high enough for phenotype alteration, supporting its immediate therapeutic application. In the present study, we have demonstrated the efficacy and potential therapeutic benefit of AAV-CRISPR/Cas9-mediated gene disruption in mouse photoreceptors. To our knowledge, this is the first report of CRISPR/Cas9-mediated gene editing in postmitotic photoreceptors *in vivo*.

We designed and evaluated a global treatment approach for rod-mediated degenerative disease by disrupting the *Nrl* gene. Indel formation was detected at the targeted *Nrl* locus in most AAV-CRISPR/Cas9-transduced cells, resulting in remarkably reduced gene expression. The transduced rods exhibited lesser expression of many rod-specific genes and enhanced expression of a few cone-specific genes compared with the WT, and displayed loss of the rod-specific chromatin pattern and diminished rod ERG response. These alterations are similar to those observed by Cre-dependent *Nrl* KO[14]. The discrepancy in expression of a few genes between the two studies may reflect the timing of intervention, percentage of *Nrl*-ablated cells, residual NRL activity and/or assays used for evaluation. Although we administered CRISPR-*Nrl* vectors at P14, 4 weeks earlier than the Cre induction in the previous study[14], we did not observe enhanced derepression of cone-specific genes, probably because of the slow onset of AAV-delivered Cas9 and sgRNA expression and delayed *Nrl* disruption. It is also likely that genomic loci of many cone genes have already acquired epigenetic marks by P14, thereby preventing their derepression by *Nrl* ablation, as demonstrated in case of *Opn1sw* gene in the previous study[14]. The CRISPR/Cas9-induced *Nrl* ablation in postmitotic rods did not seem to impact their synaptic connections with downstream neurons. Additionally, limited phenotypic alterations in transduced rods did not influence ganglion cells, retinal vasculature or RPE integrity, unlike the *Nrl* − / − mice[29]. Cone-specific visual cycle was suggested to support the phototransduction after *Nrl* ablation in mature rods[14]. Further investigations are necessary to elucidate whether the canonical visual cycle is still the major pathway of chromophore recycling for CRISPR-*Nrl*-transduced rods in the presence of an intact RPE structure.

The most important finding presented here is that *Nrl* ablation in postmitotic rods can mitigate the impact of retinal degeneration initiated by rod dysfunction and/or mutations in rod-specific genes, regardless of the pattern of inheritance. We used three different mouse models exhibiting distinct aetiology and pathophysiology of retinal degeneration—*Rho* − / −, *Rd10* and *RHO-P347S*. All three models responded favourably to the CRISPR-*Nrl* treatment as indicated by a large number of surviving rod cell bodies and relatively intact ONL. While our study does not show how *Nrl*-ablated rods can tolerate deleterious effects of disease-causing mutations, it is abundantly clear that the presence of even dysfunctional or dysmorphic rods in the ONL can protect the cone photoreceptors from secondary cell death and preserve cone-mediated vision. Many explanations can be put forward; these include coupling of rod–cone function, retention of spatial organization in ONL, continued secretion of cone survival factors by rods and reduced/lack of toxicity caused by dysfunctional/dying rods. Regardless, our results support the concept that cone death could be prevented or delayed by maintaining rod survival.

Increased survival of *Nrl*-ablated rods may be attributed to several factors, including downregulation of rod-specific genes. Mutations in genes with important photoreceptor functions often manifest their deleterious effects maximally in a fully differentiated photoreceptor environment. Several examples serve to illustrate this relationship. The *Rd1* mutation in mice (PDE6 null) leads to rapid photoreceptor loss related to greatly elevated cGMP and the ensuing increase in cation flux through cGMP-gated channels localized on the outer segment plasma membranes. When combined with the *Rds* mutation, which disrupts OS morphogenesis, cGMP levels remained elevated but the rapid photoreceptor cell death phenotype of *Rd1* mutation was 'rescued'[45], presumably because the cellular structure that enables increased cation flux is compromised. Similar examples are found in the experimental retinal detachment[46], retinal organ culture[47] and malnutrition[48] models where arrest of development or disruption to the fully differentiated state of photoreceptors are protective against mutation-induced degeneration. Interestingly, treatment with ciliary neurotrophic factor downregulates a large number of photoreceptor late genes and offers a protective effect in mutant retinas[49]. Another factor contributing to increased survival of *Nrl*-ablated rods could be related to rods being more vulnerable than cones to the deleterious effects of mutations, which has been observed in both human patients and animal models[50,51]. It is therefore conceivable that acquisition of certain cone-like traits together with the downregulation of most rod genes in *Nrl*-ablated rods promotes their survival.

In line with most preclinical gene therapy studies for inherited retinal degeneration, earlier vector administration resulted in better treatment outcomes (Fig. 6, Supplementary Fig. 18). Intervention windows in the three disease models used in the current study could be relatively narrow, due to the early onset and fast progression of the diseases, and the relatively long time course it takes for CRISPR-*Nrl* to abolish NRL expression. However, intervention window in these mice cannot be extrapolated directly to human patients. In most patients with inherited retinal diseases, noticeable loss of photoreceptors usually happens in months to years. Therefore, the time course needed for AAV-CRISPR to take effect does not seem to be critical for human application. In future studies, our approach can be tested in mouse models with relatively slow retinal degeneration with intervention given at mid- to late-stage of the diseases, better mimicking the treatment in humans. It should be noted that although our approach significantly delayed photoreceptor loss, it may not be able to completely halt disease progression. Codelivery of genes encoding neuroprotective, prosurvival or antiapoptotic factors should be considered to extend rod survival so as to improve the treatment outcome.

*In vivo* use of CRISPR/Cas9 raises two major concerns. First, *in vivo* expression of the bacterial protein Cas9 may cause immune response and other unknown adverse effects. A recent study has demonstrated cellular and humoral immune responses in mice evoked by AAV-delivered Cas9 following intramuscular administration, although significant muscle cell damage was not observed[52]. Second, CRISPR/Cas9-induced DSB may happen at off-target genome loci. These concerns are exacerbated by the persistent expression of the AAV-delivered CRISPR/Cas9 components in photoreceptors. Although little is known about the consequence of long-term Cas9 expression, no major side effects are reported in the Cas9 knock-in mouse lines[53] that have been used in a number of studies. We did not observe obvious retinal functional defect even after high-dose administration of AAV-Cas9 (Supplementary Fig. 2). Considering that retina is an immune-privileged tissue, the immune response of Cas9 may also

be alleviated. Additionally, we did not observe off-target mutations at 10 potential off-target sites even at 9.5 months post treatment (Supplementary Table 2). A controllable system that allows temporary but adequate expression of CRISPR/Cas9 components would reduce the potential side effects. As the genome editing technology is evolving rapidly, more target-specific CRISPR/Cas9 systems[54,55] will be available, which may ease the off-target concern. To improve efficiency, the current dual-vector system in which Cas9 and sgRNA are separately delivered can be replaced by a one-vector system, if SpCas9 is substituted by a shorter Cas9 from *Staphylococcus aureus*[56,57]. We believe that the continued development of this system will enhance the safety and efficacy of retinal gene editing and will find its applications in a wide range of retinal studies.

An important question from our studies is whether loss of NRL function in adult photoreceptors can be exploited for treatment of retinal degenerative diseases that are initiated in rods. The answer requires further explanation. In *Nrl*−/− mice, loss of NRL results in enhanced S-cone like phenotype[15] and, after transient degeneration, several layers of S-cone like cells survive and are functional throughout life[29]. However, gain- and loss-of-function mutations in *NRL* are responsible for diverse human retinopathies[58,59]. Unlike the rescue of retinal degeneration phenotypes by *Nrl*-ablation in mouse rods reported here, is it possible that loss of NRL in human rods will lead to cell death? We suggest that the loss of NRL in mature rods will have a distinct impact compared with what is observed by inherited *NRL* mutations. NRL is required for determining cell fate during retinal development[15]; however, unlike many differentiation factors, its high-level expression is detected throughout life in rods for maintaining rod function and homeostasis. While the former role in cell fate is unique to NRL, the latter function in maintaining rod gene expression is accomplished synergistically with CRX, NR2E3, ESRRB and other rod-expressed transcription factors[3,16,60,61] and with epigenetic controls[14,62]. Thus, as we report here, loss of NRL in adult photoreceptors is not expected to lead to change in cell fate or reprogramming; instead, the expression of many rod genes will be reduced or even abrogated in the absence of NRL, and several cone genes will likely be derepressed because of loss of NRL and dramatically decreased NR2E3 expression. We propose that these dysmorphic and dysfunctional rods can overcome rod-specific degenerative mechanisms and survive longer in diseased retina. Our studies, therefore, provide novel insights into regulation of rod photoreceptor homeostasis and suggest paradigms for treatment of retinal degenerative diseases.

## Methods

**Mouse lines and husbandry.** The C57bl/6j mice, *Nrl-L-EGFP* mice[25], *Crxp-Nrl* mice[26], *Rho*−/− mice[37], *Nrl-L-EGFP/Rd10* mice[38] and *RHO P347S* transgenic mice[39] were maintained in the National Institutes of Health (NIH) animal care facilities in controlled ambient illumination on a 12 h light/12 h dark cycle. One cohort of *Nrl-L-EGFP/Rd10* mice was dark-reared from P14 to P56 after receiving vector administration. Studies conform to ARVO statement for the Use of Animals in Ophthalmic and Vision Research. Animal protocols were approved by the National Eye Institute (NEI) Animal Care and Use Committee.

**Construction and production of AAV vectors.** The plasmid containing the cDNA of SpCas9 with N-terminal Myc tag was purchased from System Biosciences, Inc. (Palo Alto, CA, USA). For constructing the vector carrying SpCas9, the cDNA with *Sac*II and *Xho*I site was PCR amplified and placed downstream of a RK promoter[20] and upstream of a short poly-A tail (Promega, Madison, WI, USA) in an existing AAV shuttle plasmid maintained in the lab, using primers below:

SpCas9F: 5′-AGTCAGACCGCGGGCCACCATGGCTAGTATGCAGAAA-3′.
SpCas9R: 5′-AGTCACTGCTCGAGTCACTTCTTCTTCTTTGCCT-3′.

For constructing the vector carrying the sgRNA, a previously reported sgRNA scaffold with human RNA polymerase III promoter U6 (ref. 22) was synthesized and cloned into an existing AAV shuttle plasmid containing a RK-promoter-driven tdTomato expression cassette. An AAV-Null vector plasmid which contains two

AAV ITRs without any expression unit in-between was provided by Sanofi Genzyme Corporate (sequence available upon request). To generate a plasmid with a ubiquitously expressed SpCas9, the RK promoter in the above-described SpCas9 AAV shuttle plasmid was replaced by a cytomegalovirus (CMV) promoter. Similarly, to generate sgRNA plasmids with a ubiquitously expressed tdTomato, the RK promoter in the above-described sgRNA AAV shuttle plasmid was replaced by the CMV promoter.

To make AAV vectors, HEK-293 cells (ATCC CRL-1,573; ATCC, Manassas, VA, USA) grown in each roller bottle (850 cm²; Corning, New York, NY, USA) were transfected with 150 µg each of the ITR-containing vector plasmid and the two helpers encoding adenoviral components essential for AAV replication and AAV2 replication (rep)/AAV8 capsid (cap) proteins respectively, by a calcium phosphate transfection method[63]. After 48 h, the cells were collected and were disrupted by a microfluidizer (model HC 2000; Microfluidics Corporation, Newton, MA, USA). After removal of the cell debris, the cell lysate was digested with Benzonase (100 U ml⁻¹) for 1 h at 37 °C. Vector particles were then concentrated using 8% polyethylene glycol 8000. For purification, the vector-containing solution was applied to caesium chloride step gradient followed by ultracentrifugation. A second round of ultracentrifugation using a linear caesium chloride gradient was then performed to ensure vector purity. The vector band was collected with an 18-gauge needle, dialysed against Tris-buffered saline (10 mM Tris-Cl, 180 mM NaCl, pH 7.4) with 0.001% Pluronic F-68 and stored frozen at −80 °C. Titres of the vectors were determined by real-time PCR using linearized plasmid standards and primers against the RK promoter.

**Screening of gene-targeted sgRNA candidates.** HEK293 cells and HEK293-GFP cells (GenTarget, Inc. Cat. No. SC001, San Diego, CA, USA) were cultured in Dulbecco's modified Eagle's medium containing 10% fetal calf serum. To screen eight EGFP sgRNA candidates, HEK293-GFP cells were co-transfected with the plasmid containing CMV-driven SpCas9 and the plasmid containing sgRNA and CMV-driven tdTomato using Lipofectamine 2000 (Invitrogen, Carlsbad, CA, USA). Cells were collected at 48 h after transfection for SURVEYOR assay or FACS. The sorted tdTomato-positive cells were routinely cultured and passaged for 3 weeks before fluorescence microscopy, flow cytometry and SURVEYOR nuclease assay.

To screen five *Nrl* sgRNA candidates, HEK293 cells were co-transfected with the plasmids containing *Nrl* cDNA, CMV-driven SpCas9 and sgRNA together with CMV-driven tdTomato, using Lipofectamine 2000. Cells were collected at 48 h after transfection for SURVEYOR nuclease assay.

**Surveyor nuclease assay.** The amplicons were PCR-amplified using primers listed in Supplementary Tables 1 or 3 from genomic DNA of cells from different groups, and were purified using QIAquick Gel Extraction Kit (Qiagen, Hilden, Germany). The amplicons were then denatured at 95 °C and gradually reannealed to allow the formation of DNA heteroduplex. The annealed heteroduplexes were digested with SURVEYOR nuclease (Transgenomic, Inc., New Haven, CT, USA) following the manufacture's instruction. The products were visualized on a 2% (wt/vol) agarose gel. The intensity of the bands of the PCR amplicons and cleavage products were measured by using ImageJ (v1.47)[64]. The indel ratio (indel%) was calculated using equation (1)[22],

$$\text{Indel } \% = 100 \times (1 - \sqrt{1 - \frac{b+c}{a+b+c}}) \qquad (1)$$

where a is the integrated intensity of the PCR amplicon and b and c are the integrated intensities of each cleavage product.

**Subretinal injections.** Mice were anaesthetized by intraperitoneal injection of a mixture of ketamine (80 mg kg⁻¹) and xylazine (8 mg kg⁻¹) and their pupils were dilated with topical atropine (1%) and tropicamide (0.5%). Subretinal injections were performed under an ophthalmic surgical microscope. An incision was made through the cornea adjacent to the limbus at the nasal side using an 18-gauge needle. A 0.5-inch 33-gauge blunt-end needle (Hamilton 207434) fitted to a Hamilton syringe (HAM87931, 75rn) was then inserted through the incision while avoiding the lens and pushed through the retina. Each mouse received 1 µl of AAV vectors per eye. Fluorescein (100 mg ml⁻¹, AK-FLUOR, Alcon, Fort Worth, TX, USA) was included in the vector suspensions (0.1% by volume) so that the vector spread in the subretinal space can be visualized.

**Immunoblot analysis.** Nuclear protein from mouse retina was extracted using the NE-PER Nuclear and Cytoplasmic Extraction Reagents (Life Technologies, Grand Island, NY, USA) following the manufacture's instruction for immunoblot analyses of CAS9 and NRL. Total protein lysates of mouse retina were obtained using radioimmunoprecipitation assay buffer containing 1× proteinase inhibitor. Proteins were separated by SDS–PAGE and transferred to polyvinylidene difluoride membranes. The membranes were blocked with 5% nonfat dry milk for 1 h at room temperature and incubated overnight at 4 °C with the primary antibody. Then, blots were washed with Tris-buffered saline with the Tween-20 (137 mM sodium chloride, 20 mM Tris, 0.1% Tween-20, pH 7.6) for three times and incubated with the horseradish peroxidase-conjugated donkey anti-rabbit or anti-mouse secondary

antibody (Jackson Immunoresearch, West Grove, PA, USA) for 1 h at room temperature. After washing three times, the membranes were developed using SuperSignal West Pico or Femto Chemiluminescent substrate (Thermo Fisher Scientific, Rockford, IL, USA). Primary antibodies used in this study are listed in Supplementary Table 4.

**Immunofluorescence.** Mouse eyes were collected after killing. A blue tissue dye was used to mark the orientation of the eye before enucleation to ensure that all analyses were performed on equivalent areas. For fixation, eyes were immediately placed in 4% paraformaldehyde (PFA) for 1 h. The fixed tissues were soaked in 30% sucrose/phosphate-buffered saline (PBS) overnight, quickly frozen and sectioned at 10 μm thickness.

For staining, the cryosections were blocked by 5% donkey serum in PBS containing 0.1% Triton X-100 (PBST) for 1 h, and then incubated overnight at 4 °C with primary antibody diluted in 5% donkey serum. Sections were washed three times in PBST and incubated with fluorochrome-conjugated secondary antibodies and 0.2 μg ml$^{-1}$ 4,6-diamidino-2-phenylindole (DAPI) for 1 h. Sections were washed again and mounted in Fluoromount-G (SouthernBiotech, Birmingham, AL, USA). Images were captured using a confocal scanning microscope LSM700 (Zeiss, Jena, Germany).

Cell count of ONL was made along the vertical meridian at three locations to each side of the optic nerve head separated by ∼500 μm each. Each location of sections from three individual animals was imaged at ×200 magnification and cropped to a length of 300 μm. DAPI-stained cell bodies in ONL were counted using ImageJ (v1.47)[64].

For immunostaining of NRL and CRX, an alternative fixation was used. Mouse eyes were frozen immediately after collecting and then sectioned at 10 μm thickness. Brief fixation (5 min) using 1% formaldehyde was conducted before pre-adsorption in 5% donkey serum in PBST for 1 h. Antibody incubation, section mounting and imaging were conducted as described above.

Primary antibodies used in this study are listed in Supplementary Table 4. Secondary antibodies included donkey anti-rabbit, anti-mouse, anti-rat, anti-chicken antibodies conjugated with Alexa Fluor 488 or 647 (Life Technologies and Jackson Immunoresearch).

**Retinal whole-mount.** To prepare flat mount of retina, mouse eyes were enucleated as quickly as possible after killing and incubated in chilled PBS for 15 min. The orientation of the eye was indicated by intact nictitating membrane. Eye balls were then squeezed gently several times to detach the retina and fixed in 4% PFA for 1 h. The retina was separated from other parts of the eye, washed with PBST and mounted in Fluoromount-G, with photoreceptor layers facing up. Imaging was conducted on confocal scanning microscope LSM700.

**Pathological staining and electron microscopy.** Mouse eyes were fixed in PBS-buffered glutaraldehyde (2.5%)/PFA (2%) for 2 h, and dehydrated in an ethanol series (50, 70, 95 and 100%), propylene oxide and embedded in epoxy resin (Embed 812; Electron Microscopy Science, Hatfield, PA, USA) or paraffin. For Toluidine blue staining, semi-thin sections (0.5 μm) were cut and stained with Toluidine blue. For haematoxylin and eosin (H&E) staining, retinal cross-sections (5 μm) were cut and stained with H&E. H&E or Toluidine blue-stained sections were imaged with Zeiss Imager Z1 Microscope (Zeiss). For electron microscopy, ultrathin sections (100 nm) were cut, post-stained in uranyl acetate and lead citrate and observed by electron microscope.

**Electroretinogram.** A computer-based system (E2, Diagnosys LLC, Lowell, MA, USA) was used for ERG recordings in response to flashes produced with LEDs or Xenon bulbs. Mice were dark-adapted overnight. Anaesthesia and pupil dilation were conducted as described above. Corneal ERGs were recorded from both eyes using gold wire loop electrodes with a drop of 2.5% hypromellose ophthalmic demulcent solution. A gold wire loop placed in the mouth was used as reference, and a ground electrode was on the tail. For dark-adapted ERG, mice were stimulated with flashes of increasing light intensity (from 0.0001 to 1,000 cd s m$^{-2}$). Responses were computer averaged and recorded at 3 to 60 s intervals depending on the stimulus intensity. For light-adapted ERG, mice were light-adapted for 2 min, and were stimulated with flashes (from 0.3 to 100 cd s m$^{-2}$) in the presence of a white 32 cd m$^{-2}$ rod-suppressing background.

**FACS and DNA/RNA isolation.** Mouse eyes were collected after killing. Retinas were isolated and dissociated in accutase (Sigma, St Louis, MO, USA) at 37 °C for 10 min with vigorously shaking. The cell suspension was filtered using Cell Strainer (BD Biosciences, San Jose, CA, USA) and washed using Hank's balanced salt solution. Dissociated retinal cells were resuspended in Hank's balanced salt solution for FACS.

TdTomato-positive cells were collected by FACS (FACSAria; BD Biosciences). For the study of EGFP knockdown in *Nrl-L-EGFP* mice, cells with EGFP fluorescence were recorded, but not necessarily collected. DNA was isolated and purified from collected cells using DNeasy Blood & Tissue Kit (Qiagen) following the manufacturer's instruction. For RNA isolation, cells were collected and

dissolved in Trizol LS reagent (Invitrogen). RNA was isolated following the manufacturer's instruction.

**Targeted deep DNA sequencing.** Candidate off-target sites were predicted using the CRISPR Design tool (http://crispr.mit.edu/)[22]. The fragments containing the on-target site of *Nrl* gene and 10 potential off-target sites were PCR-amplified using the primers listed in Supplementary Table 2. The amplicons were gel purified using QIAquick Gel Extraction Kit, quantified using Nanodrop (Thermo Fisher Scientific, Waltham, MA, USA) and pooled in equal ratio. An aliquot of the pool was run on an Illumina MiSeq using a MiSeq Reagent Nano kit, ver2 (Illumina, San Diego, CA, USA). This quality control run consisted of 25 cycles followed by two index reads. The pool was then rebalanced based on the percentage of reads seen for each amplicon's indexes. The final pool is then sequenced on the MiSeq using MiSeq Reagent Kit ver3 (Illumina) to generate paired-end 300 base reads. Post-run processing of data is performed using RTA 1.18.54 and CASAVA (v1.8.2.). Aligned Binary Alignment/Map (BAM) files were converted to FASTQ format using the Bedtools software package[65] and analysed using the CRISPResso software package[66].

**RNA sequencing and data analysis.** Total RNA of flow-sorted tdTomato-positive cells was isolated using TRIzol LS (Invitrogen). RNA quality was assessed using Bioanalyzer RNA 6000 Pico assay (Agilent Technologies, CA, USA). Strand-specific mRNA sequencing libraries were generated from 20 ng of total RNA using TruSeq RNA Sample Prep Kit-v2 (Illumina)[67] and sequenced on Hi-Seq 2000 (Illumina).

Qualitative assessment of the FASTQ files was evaluated using FastQC (http://www.bioinformatics.babraham.ac.uk/projects/fastqc/). The reference assembly GRCm38.p3 with gene annotation Ensemble 84 was used for alignment and quantification. STAR[68] software package was utilized for alignment and gene-level quantification. ENCODE guidelines for parameter settings for STAR were used and gene level counts were obtained by setting --quantMode parameter to 'GeneCounts'. Gene-level counts were used as input to edgeR[69] package for computing count per million (CPM) values and to normalize the data via library size correction using the TMM normalization method. Limma[70] package was used to perform differential expression analysis that yielded the fold-change values and its associated *P* values. All packages used were developed and implemented in R programming language.

To identify enriched GO terms for DE genes, we used GOrilla web tool (http://cbl-gorilla.cs.technion.ac.il)[31,32]. We chose two unranked list of genes option and provided list of DE genes and list of genes that were expressed ≥1 CPM in control (CRISPR-EGFP treated) or CRISPR-*Nrl*-treated data sets. For the ontology option we chose 'Process' option, and all other options were set to default values.

**Optomotor test.** Optomotor responses were tested using the OptoMotry system (Cerebral Mechanics, Inc., Lethbridge, Canada). Mouse was placed in the centre of a closed chamber surrounded by four computer monitors with a camera on the top to monitor the action of the animals. Three-dimensional virtual images of rotating drum lined with vertical sine wave grating were projected on the monitors. The spatial frequency of the grating was controlled by OptoMotry software (Ver14). Tracking of the gratings by the mouse was reflected by its head and neck movement. The maximum spatial frequency (in cycles/degree) of each eye was measured by systematically increasing the spatial frequency of the grating at 100% contrast until the mouse no longer tracks.

**Fundus fluorescein angiography.** Fundus angiography of mice was imaged using Micron III Rodent Fundus Imaging System (Phoenix Research Labs, Pleasanton, CA, USA), equipped with a 390–490 nm excitation filter and a 500 nm long-pass emission filter. Mice were anaesthetized and their pupils were dilated as described above. The mice were then intraperitoneally injected with 10% of sodium fluorescein at a dose of 0.02 ml per 5 g body weight. Fundus images were taken at 5 min after injection.

**Statistical analysis.** A two-tailed paired *t*-test was used to determine the statistical significance between the eyes receiving the treatment vectors and the eyes receiving the control vectors. Statistical analyses were performed using GraphPad Prism 6 (GraphPad Software, La Jolla, CA, USA).

**Data availability.** The data that support the findings of this study are available from the corresponding author upon reasonable request.

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

## Acknowledgements

We thank Wei Li, Jacob Nellissery, Jessica Gumerson, Yide Mi, Megan Kopera and Linn Gieser for their assistance. We also thank the Visual Function Core, Flow Cytometry Core, Histopathology Core and Biological Imaging Core of the National Eye Institute and the NIH Intramural Sequencing Center (NISC) for their assistance. This work was supported by the intramural research program of the National Eye Institute.

## Author contributions

W.Y. and Z.W. designed research; W.Y., S.M., V.C., S.H., J.-W.K., M.B., Y.A., X.S. and Z.W. performed research; W.Y., V.C., J.-W.K., M.B., L.D., T.L., A.S. and Z.W. analysed data; W.Y. and Z.W. wrote the manuscript; T.L., A.S. and Z.W. edited the manuscript.

## Additional information

**Competing financial interests:** The authors declare no competing financial interests.

