## [Peer review file · Nature Communications]

Reviewers' comments:

Reviewer #1 (Remarks to the Author):

The authors demonstrated cone degeneration could be slowed down by Nrl knockdown via AAV-delivered CRISPR/Cas9 system in autosomal dominant and recessive models for RP. The study is well designed; results are well presented. However, the concept is not novel, since it was reported before with different approaches (Montana et. al., 2013, PMID: 23319618). The major concerns for potential clinical application as global approach to treat gene-independent RP are: A central role for rods is retinal homeostasis. In human retina, rod photoreceptors out-number cone photoreceptors (about 70% of retinal cells) and the consequent secondary and tertiary neurons recruited for processing rod-drive vision outnumber the cone pathways neurons everywhere except in the central fovea. Rods form hexagonal packing around the cones and separate the cones from each other. Loss of rods has a major influence on retinal architecture. Study has shown that loss of rods compromises the fragile homeostasis in the retina of Nrl^{-/-} mice, which results in Müller and microglia cell activation, altering retinal vasculature, loss or dysfunction of retinal neurons and RPE atrophy. With the rapid development of gene editing technology, design of gRNAs based on each individual patient's specific mutation should be feasible.

Here are comments for further consideration:

1. Most injections were performed at P14 when rods are not mature yet. It will be more important if the intervention was done at late time point when degeneration has started. Author did performed one injection at P28 when rods just started to degenerate in Rho^{-/-} mice. The outcomes are not as good as that from early intervention. Can authors comment on the intervention window? What results do authors expect if intervention is performed when half of the rods have gone?

2. Authors checked two time points on ERG in RHO P347S mice, the b wave amplitude declined fast within 3 weeks (from P67 to P90). This raises concerns the long-term effect after Nrl ablation.

3. Authors did not examine the outcomes on retinal secondary, tertiary neurons, synapses and transmitters after Nrl ablation. Structural changes including dismantling of synapses, degradation of circuitry and rewiring have been well studied in neurodegenerative diseases. Retina is highly connected and interacted tissue, loss of rods morphologically and functionally may affect neurons that have direct connection with, for examples, there is diversity of human retinal bipolar types, some postsynapse only to rods, some to cones and still others receive mixed rod-cone input; Humans with ON-bipolar deficit lost nocturnal vision and suffered from reduced sensitivity of cone vision.

4. Absence of rod photoreceptors or secretion of a toxic product by excess cones appears to compromise the integrity of RPE. The integrity of both RPE and vascular structure should be examined. Authors should make comments on visual cycle changes after Nrl ablation, such as, were rod outer segments still recycled by RPE cells or Müller glia?

Other minor suggestions:

1. How the ONL thickness was measured and selected? Please provide detail in the method section.

2. Line 343: All the experiments except one were performed at P14, when rods are not matured. Suggest to change mature to post-mitotic rods.

3. Figure 7 f: DAPI stained sections, INL in CRISPR-EGFP treated retina seems to be much thicker, compared with the CRISPR-Nrl treated retina.

Reviewer #2 (Remarks to the Author):

***Summary of the key results**

The study by Yu, et al. performed a POC study of AAV-based gene therapy to protect rod photoreceptor degeneration. They constructed AAVs carrying SpCas9 and sgRNA for Nrl, a rod photoreceptor-specific transcription factor, and introduced the AAVs in mouse to cause a mutation in the Nrl loci in the developed rod photoreceptor cells, which resulted in the down-regulation of rod photoreceptor-specific genes and upregulation of cone photoreceptor-specific genes. They performed the experiment using three rod degeneration mouse lines, and showed that AAV injection before rod degeneration protected the rod photoreceptors from cell death and preserved visual responses.

***Originality and interest: if not novel, please give references**

As the authors mentioned, this study was based on a previous report by Montana et al. (PNAS 110:1732-7. 2013) showing that conditional mutagenesis of the Nrl gene loci in the developed Rho^{-/-} rod photoreceptors delayed rod photoreceptor degeneration. While Montana et al. disrupted Nrl using a Cre-loxP system, the authors in this study used an AAV and Crispr system. Notably, they demonstrated that this experimental idea could be applied to other rod degeneration mice with different causal genes. When considering that many causal genes have been reported for human retinitis pigmentosa, their results hold promise for practical gene therapy.

***Data & methodology: validity of approach, quality of data, quality of presentation**

***Conclusions: robustness, validity, reliability**

They have clearly presented the developmental characterization of Nrl-ablated rods. However, they have not examined any molecules involved in protection against rod photoreceptor degeneration. A number of researchers have been studying cytokine pathway genes to protect photoreceptor degeneration. Previous transcriptome analyses (GEO profiles: GSE33141, GDS1693, GDS2936) have shown upregulation of several cytokine pathway genes (Cntf, Osmr, Stat3, Gp130, Socs3) in the adult Nrl KO mouse retinas, and rod-specific overexpression of Stat3 exhibited delayed rod degeneration in RHO-P347S mice (PNAS 111:E5716-23. 2014). These previous studies suggest that cytokine pathway genes were upregulated in the Nrl-ablated rods in this study.

***Suggested improvements: experiments, data for possible revision**

***Appropriate use of statistics and treatment of uncertainties**

Line 144 (Fig 1a). Please describe the vector construction of "AAV-Null". Does the AAV carry a RK promoter and a short polyA?

Line 146 (Fig 1e, f, g). Nrl-L-EGFP mice were reported to have three copies of the transgene. Did the authors observe any differences in GFP intensity in the EGFP+/TdTomato+ (yellow) fraction? It is also not clear whether AAV-sgRNA-EGFP and AAV-Cas9 were co-transfected into a single cell. Is it possible to test section IHC with anti-myc antibodies to visualize Cas9-infected cells?

Line 197 (Fig 3b). There appears to be fewer cone-like nuclei compared with photoreceptor nuclei in Acute Nrl KO by Montana et. al. (PNAS 110:1732-7. 2013) despite earlier disruption of Nrl in this study. Is this because Crxp-Nrl transgenic mice were used?

Line 215 (Fig 4a). Grk1 (rhodopsin kinase) is expressed in both rod and cone photoreceptors. Please address this in the figure or in the text.

Figs 5-7. The authors showed optomotor data only in RHO-P347S mice (Fig 7e). Have they tested optokinetic response in Nrl-disrupted Rho^{-/-} and Rh10 mice (Figs 5 and 6)?

Reviewer #3 (Remarks to the Author):

The authors develop a CRISPR Cas9 gene editing approach for retinal degeneration diseases. Specifically they target Nrl and show a rescue effect in 3 models.

- Concerns exist with regard to how long Cas9 remains using AAV. The authors should check for long term Cas9 expression and potential for off target at later time points.

- Is the gain in cone features maintained over time?

- Can the authors demonstrate that no other cell types are transduced? Difficult to see specificity of targeting in Fig 1g.

- The small changes seen in gene expression is less than predicted. This should be more clearly evaluated/discussed.

- The authors should include an epigenetic evaluation of changes in cell fate to further demonstrate acquisition of cone features.

- Is the heterogeneity seen due to global inefficiencies in targeting or clone differences upon targeting NRL? What is leading to the variability? Do you see lower or higher changes among different clones. If targeting is higher or lower in different clones does this correlate with changes in gene expression or do you see higher or lower. This would assist in directly correlating targeting to the cell fate and gene expression changes.

- Can you molecularly or immunocytochemically demonstrate changes in survival or cell death genes in NRL targeted cells?

- Is targeting one or both alleles efficiently or does this vary per cell?

Point-to-point response

[Reviewer #1] The authors demonstrated cone degeneration could be slowed down by *Nrl* knockdown via AAV-delivered CRISPR/Cas9 system in autosomal dominant and recessive models for RP. The study is well designed; results are well presented.

We appreciate the reviewer's comments.

[Reviewer #1] However, the concept is not novel, since it was reported before with different approaches (Montana et al., 2013, PMID: 23319618). The major concerns for potential clinical application as global approach to treat gene-independent RP are: A central role for rods is retinal homeostasis. In human retina, rod photoreceptors outnumber cone photoreceptors (about 70% of retinal cells) and the consequent secondary and tertiary neurons recruited for processing rod-drive vision outnumber the cone pathways neurons everywhere except in the central fovea. Rods form hexagonal packing around the cones and separate the cones from each other. Loss of rods has a major influence on retinal architecture. Study has shown that loss of rods compromises the fragile homeostasis in the retina of *Nrl*^{-/-} mice, which results in Müller and microglia cell activation, altering retinal vasculature, loss or dysfunction of retinal neurons and RPE atrophy.

We completely understand the reviewer's concerns regarding the potential consequences of altered retinal homeostasis following AAV-CRISPR mediated *Nrl* knockdown. As the reviewer pointed out, in germline *Nrl* knockout mice, conversion of rods to S cones compromises the homeostasis of the retina, resulting in Müller/microglia cell activation, altered retinal vasculature, loss or dysfunction of retinal neurons and RPE atrophy, as reported previously (Roger et al, 2013, J Neuroscience, 32:528). However, in the present approach, AAV-CRISPR against *Nrl* was administered at postnatal day 14 when rods were postmitotic and nearly mature. Considering that a few days to a few weeks are needed from AAV vector administration to *Nrl* disruption, complete *Nrl* ablation in transduced rods may take place when they are fully mature. Due to the lack of plasticity at this stage, the *Nrl*-ablated rods were not able to be reprogrammed to bona fide cones. A strong evidence supporting this is the global transcriptome analysis (Supplementary Fig. 11) revealing only 146 differentially expressed genes in the CRISPR-*Nrl* treated postmitotic photoreceptors, in contrast to the 6412 genes differentially expressed between mature rods and S cone-like cells (Kim et al, 2016, Dev Cell, 37:520). Additionally, substantial residual rod-mediated retinal function was preserved after CRISPR-*Nrl* treatment, as detected by dark-adapted ERG (Fig. 3c,d). Morphologically, only a portion of rod nuclei obtained cone-like features (Fig. 3a,b). We do not think these can be simply interpreted as the results of low transduction of CRISPR-*Nrl* vectors or suboptimal inactivation of *Nrl* gene, as *Nrl* ablation appeared to have happened in most transduced cells, as revealed by the DNA deep sequencing, immunoblot, and immunofluorescence analyses (Fig. 2). We therefore believe that the *Nrl*-ablated rods, though gained certain cone features, only moderately altered their rod phenotypes, which did not significantly affect the retinal homeostasis. We followed the reviewer's suggestions and performed additional experiments to examine Müller cell activation, retinal vasculature, inner retinal

structure and RPE integrity (Supplementary Fig.8, 9, 10), but did not observe significant changes following CRISPR-*Nrl* treatment (see below for detailed discussion).

[Reviewer #1] With the rapid development of gene editing technology, design of gRNAs based on each individual patient's specific mutation should be feasible.

We agree with the reviewer that design of gRNA against each specific gene mutation will not be a problem with the ever-increasing number of naturally occurring or engineered CRISPR systems. However, the CRISPR system only creates double-strand DNA break (DSB) at the mutation locus, which is the first step toward an efficient gene modification. A precise gene repair relies on the homologous recombination (HR) occurring between the repair template and the mutation locus, which is extremely inefficient in post-mitotic photoreceptors. Progress has been made recently toward improving HR efficiency (Richardson *et al*, 2016, *Nat Biotechnol*, 34:339; Maruyama *et al*, 2015, *Nat Biotechnol*, 33:538). However, these approaches appear to be far from direct *in vivo* application. Although precise repair/modification of gene mutations *in vivo* for inherited retinal degeneration could be achieved in the future, developing therapies based on CRISPR's gene disruption function relying on the more efficient non-homologous end joining (NHEJ) pathway, could be a more practical approach at the current stage.

[Reviewer #1] Here are comments for further consideration:

1. Most injections were performed at P14 when rods are not mature yet. It will be more important if the intervention was done at late time point when degeneration has started. Author did performed one injection at P28 when rods just started to degenerate in *Rho*^{-/-} mice. The outcomes are not as good as that from early intervention. Can authors comment on the intervention window? What results do authors expect if intervention is performed when half of the rods have gone?

We thank the reviewer's questions pertaining to window of gene therapy intervention for retinal degeneration using AAV vectors. In a majority of successful pre-clinical studies in animal models of retinal degeneration, AAV vectors were given prior to the onset of photoreceptor death. This held true especially in models with rapid degeneration (Tan *et al*, 2009, *Hum Mol Gen*, 18:2099; Sun *et al*, 2010, *Gene Ther*, 17:117). Intervention window in each disease model is determined by multiple factors, including the onset and speed of photoreceptor death, and the onset and strength of the vector-mediated gene expression. In *Rho*^{-/-} mice, rod death starts at around postnatal day 24 (P24) (Humphries *et al*, 1997, *Nat Genet*, 15:216), and complete rod death occurs at around P90 (Lem J *et al*, 1999, *Proc Natl Acad Sci U S A*, 96:736). One of the main reasons for early intervention relates to the biology of AAV, a single-stranded DNA vector packaged with a protein capsid, which undergoes a relatively slow intracellular process including a rate-limiting uncoating step (Thomas *et al*, 2004, *J Virol*, 78:3110) before a transcriptionally competent double-stranded DNA is formed. It has been well-established in gene replacement (or gene augmentation) studies that a few days to a few weeks are needed for the AAV-delivered genes to reach therapeutic levels following vector administration. In our current approach, it may take an even longer time period for the NRL abolishment to happen in a majority of cells, as expression of the delivered Cas9 and gRNA is only half

way toward *Nrl* ablation. Additional steps including formation of Cas9-gRNA complex, target sequence search, generation of double-strand DNA break (DSB), and repair of DSB through the NHEJ pathway, also take time. Furthermore, degradation of residual NRL protein needs to be taken into account as well. Our EGFP targeting experiment revealed a better EGFP knockdown at 10-week post treatment than that at 6-week post treatment (Fig. 1e), supporting a long time course needed for AAV-CRISPR mediated abolishment of expression. This may interpret why intervention at P14 is better than that at P28. As a proof-of-concept for *in vivo* CRISPR/Cas9 application in the retina, our current study did not include intervention at mid- or late-stage of the diseases. If intervention is performed when half of the rods have degenerated (around P50), we speculate that only a minimal therapeutic effect could be achieved, since only a small number of rods would still be alive when ablation of *Nrl* is complete.

However, intervention window in mouse models with rapid photoreceptor degeneration cannot be extrapolated directly to human patients. In most patients with inherited retinal diseases, noticeable loss of photoreceptors usually happens in months to years, which is much slower than that in mice with rapid degeneration. Therefore, the time course needed for AAV-CRISPR to take effect does not seem to be critical for human application. If intervention is provided to human patients when half of the rods have gone, we may expect a better treatment outcome than that in mouse models. In future studies, our approach could be tested in mouse models with relatively slow retinal degeneration (such as *rds* mice) with intervention given at mid- or late-stage of the diseases, which could better mimic the treatment in humans.

We have added a new paragraph to Discussion regarding the intervention window.

“In line with most preclinical gene therapy studies for inherited retinal degeneration, earlier CRISPR-*Nrl* vector administration resulted in better treatment outcomes. Intervention windows in the three disease models used in the current study could be relatively narrow, due to the early onset and fast progression of the diseases, and the relatively long time course it may take for CRISPR-*Nrl* vectors to abolish NRL expression. However, intervention window in mice with rapid photoreceptor degeneration cannot be extrapolated directly to human patients. In most patients with inherited retinal diseases, noticeable loss of photoreceptors usually happens in months to years. Therefore, the time course needed for AAV-CRISPR to take effect does not seem to be critical for human application. In future studies, our approach could be tested in mouse models with relatively slow retinal degeneration with intervention given at mid- to late-stage of the diseases, which could better mimic the treatment in humans”.

[Reviewer #1] 2. Authors checked two time points on ERG in RHO P347S mice, the b wave amplitude declined fast within 3 weeks (from P67 to P90). This raises concerns the long-term effect after *Nrl* ablation.

We understand the reviewer’s concern. Our original plan was to examine whether AAV-CRISPR mediated *Nrl* knockdown could significantly delay retinal degeneration and preserve visual function in the disease models. As all three mouse models exhibit severe

photoreceptor degeneration at post-natal day 90, we thought that a 90-day monitoring could be sufficient to draw a conclusion. This has been confirmed by our ERG and retinal morphology results (Figures 5 to 7). In the revised version of our manuscript, we have included the 4-month ERG and retinal morphology results collected from 6 additional mice receiving vector treatment (Supplementary Fig. 19). Each individual mouse still displayed higher photopic ERG amplitude in the CRISPR-*Nrl* treated eye than in the control eye, indicating that the therapeutic effect persisted at least to 4 months. A longer term of monitoring beyond 4 months was not pursued, as it was not included in our original plan.

Although our approach can remarkably delay rod degeneration and cone function loss, it may not be able to completely halt disease progression. This appears to be a common problem in gene therapy studies for many inherited retinal degenerations and has been observed both in animal models and in clinical trials in patients (Cideciyan *et al.* 2013, Proc Natl Acad Sci U S A. 110:E517). It was suggested that there is a threshold of accumulated molecular changes as a result of gene mutation, after which photoreceptor death is inevitable, although this was challenged by a recent study showing that even in advanced stage of retinal degeneration, gene replacement could still be able to halt the disease (Koch *et al.*, 2015, J Clin Invest. 125:3704). Insufficient or suboptimal retinal transduction by AAV vectors could be an alternative interpretation, as it allows abnormalities in un-transduced mutant cells to continue to stress both transduced and un-transduced photoreceptors (Koch *et al.* 2015, J Clin Invest. 125:3704). However, even in ideal situation where intervention is early enough and vector transduction is sufficient, it is still possible that the *Nrl*-ablated rods are unable to fully resist the effects of rod gene mutations and would gradually degenerate. Although the approach is not perfect, the therapeutic effects were relatively long-lasting (at least 4 months, Supplementary Fig. 19), which is comparable or better than a number of therapy studies conducted on the same disease models using gene replacement or other protection strategies (*Rd10* model: Allocca *et al.*, 2011, Invest Ophthalmol Vis Sci, 52:5713; Yao *et al.*, 2012, PLoS One, 7:e37197; Pang *et al.*, 2008, Invest Ophthalmol Vis Sci. 49:4278. *RHO P347S* model: Jiang *et al.*, 2014, Proc Natl Acad Sci U S A, 111:E5716; Millington-Ward *et al.*, 2011, Mol Ther, 19:642; Bramall *et al.*, 2013, PLoS One, 8:e58023). We believe that our approach deserves pursuing as a global treatment for retinal degeneration caused by various gene mutations. In future studies, co-delivery of genes encoding neuroprotective, prosurvival, or antiapoptotic factors should be considered to extend rod survival so as to improve the outcome of this approach.

We added the following sentences to address the reviewer's concern.

“It should be noted that although our current approach significantly delayed photoreceptor loss, it may not be able to completely halt disease progression. Co-delivery of genes encoding neuroprotective, prosurvival, or antiapoptotic factors should be considered to extend rod survival so as to improve the treatment outcome”.

[Reviewer #1] 3. Authors did not examine the outcomes on retinal secondary, tertiary neurons, synapsis and transmitters after *Nrl* ablation. Structural changes including

dismantling of synapses, degradation of circuitry and rewiring have been well studied in neurodegenerative diseases. Retina is highly connected and interacted tissue, loss of rods morphologically and functionally may affect neurons that have direct connection with, for examples, there is diversity of human retinal bipolar types, some postsynapse only to rods, some to cones and still others receive mixed rod-cone input; Humans with ON-bipolar deficit lost nocturnal vision and suffered from reduced sensitivity of cone vision.

We have performed a few additional experiments to address the reviewer's concerns. These experiments included PKC α , mGluR6, BRN3A and NF-L staining to examine changes on bipolar cells, bipolar synapses and ganglion cells, respectively at 3 and 7 months of age after vector administration. No remarkable differences were observed between the CRISPR-*Nrl* vector treated retina and the control retina at both time points, indicating that *Nrl* ablation does not significantly affect the retinal secondary, tertiary neurons and synapsis (Supplementary Fig. 8). This is not unexpected, as even in germline *Nrl* knockout retina, rod bipolar cells have normal morphology, pattern of staining and lamination, although they form synaptic connections with S cones (Strettoi *et al*, 2004, J Neurosci, 24:7576). As mentioned earlier, in the present study, *Nrl* ablation only moderately altered rod phenotypes, which does not seem to significantly affect the connection with downstream neurons. In germline *Nrl* KO retina, prominent ganglion cell (GC) death was observed, probably resulting from the rapid yet transient cone cell death between 1 and 4 months of age and the subsequent retinal vasculature alteration and optic atrophy (Roger *et al*, 2013, J Neuroscience, 32:528). However, in the present study we did not observe obvious changes in the retinal vasculature at ~7 months after CRISPR-*Nrl* treatment (Supplementary Fig. 10). GC death at 7 months of age was not obvious either (Supplementary Fig. 8).

[Reviewer #1] 4. Absence of rod photoreceptors or secretion of a toxic product by excess cones appears to compromise the integrity of RPE. The integrity of both RPE and vascular structure should be examined.

We followed the reviewer's suggestions and performed additional experiments. In contrast to *Nrl* KO retina which exhibited compromised RPE integrity and abnormal structure and permeability of retinal vasculature (Roger *et al*, 2013, J Neuroscience, 32:528), no obvious changes were observed in both RPE and retinal vasculature in C57/Bl6 mice at ~7 months of age after CRISPR/*Nrl* treatment, using Ezrin and RPE65 staining, and fluorescein angiography (Supplementary Fig. 9, 10). We also examined Muller cells by Sox9 and GFAP staining. Unlike *Nrl* KO retina (Roger *et al*, 2013, J Neuroscience, 32:528), no displacement of Muller cell nuclei was found after CRISPR-*Nrl* treatment (Supplementary Fig. 9). Only mild gliosis was seen in both CRISPR-EGFP and CRISPR-*Nrl* treated retina but not in untreated retina (Supplementary Fig. 9), indicating the response to stress probably induced by CRISPR/Cas9 mediated reaction.

To address question # 3 and #4 from the reviewer, we have added the following to Results.

“No obvious differences were observed between CRISPR-*Nrl* treated and control retinas in bipolar cells and post-photoreceptor synapses, similar to previous findings in *Nrl*^{-/-} mice. Location of retinal ganglion cells, integrity of RPE and retinal vasculature were not noticeably altered, in contrast to those observed in *Nrl*^{-/-} mice. Mild gliosis was observed in both CRISPR-EGFP and CRISPR-*Nrl* treated retinas, indicating stress response caused by Cas9 expression and/or DSB creation and repair. However, location of Muller cell nuclei was not altered, different from that of *Nrl*^{-/-} retina. These results collectively indicated that acquisition of cone-like morphology in some rods after CRISPR-*Nrl* treatment does not significantly alter the overall retinal structure.”

“Similar to those observed in germline *Nrl*^{-/-} mice, the CRISPR/Cas9 induced *Nrl* ablation in postmitotic rods did not seem to affect their connections with downstream neurons. Unlike *Nrl*^{-/-} mice, ganglion cell death, compromised retinal vasculature and RPE integrity were not seen either, probably due to the limited phenotype alteration in transduced rods.”

[Reviewer #1] Authors should make comments on visual cycle changes after *Nrl* ablation, such as, were rod outer segments still recycled by RPE cells or Müller glia?

In addition to the canonical RPE visual cycle pathway, a cone-specific visual cycle mainly involving Muller cells was suggested and has been gaining acceptance in recent years (Wang *et al*, 2009, Nature Neurosci, 12: 295; Xue *et al*, 2015, J Clin Invest, 125:727). The previous study has shown that in isolated retina free of RPE, the *Nrl*-ablated rods were able to use 9-cis-retinol for recovery of photoresponse (Montana *et al*, 2013, Proc Natl Acad Sci U S A, 110:1732), suggesting that these cells could use the cone-specific visual cycle as their chromophore recycling pathway. Although we favor this conclusion, we did not perform the same experiment using an isolated CRISPR-*Nrl* treated retina. Additionally, it is unclear in the presence of an intact RPE structure *in vivo*, whether the CRISPR-*Nrl*-treated rods use the cone-specific visual cycle as their major pathway of chromophore recycling, as the capacity of chromophore recycling of Muller cells is limited. We added the following in Discussion.

“It was suggested that adult rods with induced *Nrl* ablation gained the ability to use the cone-specific visual cycle to support their functions. However, it remains unclear in the presence of an uncompromised RPE structure whether the canonical visual cycle is still the major pathway of chromophore recycling for CRISPR-*Nrl* transduced rods.”

[Reviewer #1] Other minor suggestions:

1. How the ONL thickness was measured and selected? Please provide detail in the method section.

We have provided the details in Materials and Methods.

2. Line 343: All the experiments except one were performed at P14, when rods are not matured. Suggest to change mature to post-mitotic rods.

We have made the corrections according to the reviewer's suggestion.

3. Figure 7 f: DAPI stained sections, INL in CRISPR-EGFP treated retina seems to be much thicker, compared with the CRISPR-Nrl treated retina.

We examined additional retinal sections and found that the one we have shown was not representative. We apologize for this oversight and have provided a new one in the current version.

Reviewer #2 (Remarks to the Author):

*Summary of the key results

The study by Yu, et al. performed a POC study of AAV-based gene therapy to protect rod photoreceptor degeneration. They constructed AAVs carrying SpCas9 and sgRNA for Nrl, a rod photoreceptor-specific transcription factor, and introduced the AAVs in mouse to cause a mutation in the Nrl loci in the developed rod photoreceptor cells, which resulted in the down-regulation of rod photoreceptor-specific genes and upregulation of cone photoreceptor-specific genes. They performed the experiment using three rod degeneration mouse lines, and showed that AAV injection before rod degeneration protected the rod photoreceptors from cell death and preserved visual responses.

*Originality and interest: if not novel, please give references

As the authors mentioned, this study was based on a previous report by Montana et al. (PNAS 110:1732-7. 2013) showing that conditional mutagenesis of the Nrl gene loci in the developed Rho^{-/-} rod photoreceptors delayed rod photoreceptor degeneration. While Montana et al. disrupted Nrl using a Cre-loxP system, the authors in this study used an AAV and Crispr system. Notably, they demonstrated that this experimental idea could be applied to other rod degeneration mice with different causal genes. When considering that many causal genes have been reported for human retinitis pigmentosa, their results hold promise for practical gene therapy.

We appreciate the reviewer's comments.

*Data & methodology: validity of approach, quality of data, quality of presentation

*Conclusions: robustness, validity, reliability

They have clearly presented the developmental characterization of Nrl-ablated rods. However, they have not examined any molecules involved in protection against rod photoreceptor degeneration. A number of researchers have been studying cytokine pathway genes to protect photoreceptor degeneration. Previous transcriptome analyses (GEO profiles: GSE33141, GDS1693, GDS2936) have shown upregulation of several cytokine pathway genes (Cntf, Osmr, Stat3, Gp130, Socs3) in the adult Nrl KO mouse retinas, and rod-specific overexpression of Stat3 exhibited delayed rod degeneration in RHO-P347S mice (PNAS 111:E5716-23. 2014). These previous studies suggest that cytokine pathway genes were upregulated in the Nrl-ablated rods in this study.

The reviewer has correctly mentioned that several cytokine related genes were differentially expressed in previously published transcriptome analyses comparing WT and *Nrl*^{-/-} retina. Though several retina specific data sets are available, we preferred to use the most recently published flow sorted rod photoreceptors (FSPR) and S-cone like (FSPR NRL-KO) data sets (Kim *et al*, 2016, *Dev Cell*, 37:520) for comparative analysis. RNA-seq analysis pipeline (see methods) for expression quantification and differential expression (DE) analysis at P28 (adult stage) yielded 6412 genes that were differentially expressed in WT vs *Nrl*^{-/-} data sets. Performing gene ontology (GO) analysis using GOrilla web tool and pathway enrichment analysis using DAVID web tool did not yield any enriched clusters or pathways related to cytokine pathway. This analysis suggests that though some cytokine pathway related genes were differentially expressed, no significant group of cytokine pathway related genes were enriched or over-represented in the DE set of genes comparing WT rods with S cone-like cells.

We also re-analyzed our transcriptome data collected from flow-sorted tdTomato-positive C57/Bl6 photoreceptors following CRISPR-*Nrl* treatment (Supplementary Fig. 11). GO enrichment analysis of the differentially expressed 146 genes following CRISPR-*Nrl* treatment revealed 20 enriched processes (Supplementary Fig. 11b). However, no cytokine pathways were enriched by the analysis. The significance of the 20 enriched processes requires further investigation, but none of them seemed to be directly related to rod protection. Currently we do not have enough evidence to show the involvement of certain molecules or pathways for rod protection.

*Suggested improvements: experiments, data for possible revision

*Appropriate use of statistics and treatment of uncertainties

Line 144 (Fig 1a). Please describe the vector construction of "AAV-Null". Does the AAV carry a RK promoter and a short polyA?

We thank the reviewer's suggestion. The AAV-Null vector plasmid was provided by Genzyme, which does not carry any promoter for gene expression. We have now provided the information in Materials and Methods.

Line 146 (Fig 1e, f, g). *Nrl*-L-EGFP mice were reported to have three copies of the transgene. Did the authors observe any differences in GFP intensity in the EGFP⁺/TdTomato⁺ (yellow) fraction? It is also not clear whether AAV-sgRNA-EGFP and AAV-Cas9 were co-transfected into a single cell. Is it possible to test section IHC with anti-myc antibodies to visualize Cas9-infected cells?

We did observe a wide spectrum of EGFP intensity in the EGFP⁺/tdTomato⁺ fraction of cells, as revealed by the FACS analysis (Fig. 1e the lowest panel). The high EGFP intensity in a majority of this cell population could mainly result from lack of Cas9 expression and failure of EGFP targeting, while the cells with lower EGFP intensity might reflect the incomplete EGFP ablation due to multiple copies of EGFP genes and/or generation of inframe indels unable to abolish EGFP expression. We have added the following description in the text.

“Lack of successful EGFP ablation in the rest 57% of the sgRNA-transduced cells could be caused by lack of Cas9 expression, in-frame indels unable to abolish EGFP expression and multiple copies of EGFP transgene in the mice (Ref) that have exceeded the capacity of CRISPR-mediated gene disruption”.

We tried a few times for Cas9 immunostaining but neither an anti-myc antibody nor an anti-Cas9 antibody worked well on photoreceptor nuclei. A previous report revealed over 50% co-transduction of photoreceptors receiving administration of two AAV5 vectors at 1:1 ratio, with a dose of 1.5×10^9 vg of each vector (Palfi *et al.*, 2012, Hum Gene Ther, 23:847). In our study, a higher dose of two AAV8 vectors (2.5×10^9 vg each) was used for Cas9 and sgRNA-EGFP co-delivery. AAV8 is more efficient than AAV5 for photoreceptor transduction (Allocca *et al.* 2007, J Virol, 81:11372). We therefore believe that a majority of transduced photoreceptors were co-transduced with both the Cas9 and the sgRNA-EGFP vectors.

We were using a different vector ratio for CRISPR-*Nrl* delivery (5×10^9 vg for Cas9, 2.5×10^9 vg for sgRNA-*Nrl*). To evaluate the efficiency of co-transduction at this ratio, we co-injected AAV8-RK-EGFP (5×10^9 vg) and AAV8-sgRNA (with RK-tdTomato) (2.5×10^9 vg) vectors subretinally, and conducted FACS analysis at 1-month post injection. As shown in supplementary Fig. 5, co-transduced cells accounted for ~76% of total transduced cells and ~89% of tdTomato-transduced cells. Although the EGFP vector dose was 2-fold higher, this result could still support a high likelihood of co-transduction by two vectors in a single cell using 1:1 vector ratio. We have added the following to Results.

“Co-transduction of two AAV vectors was observed in a majority of photoreceptors following subretinal delivery of two reporter AAV vectors using the same ratio and doses (Supplementary Fig. 5).”

Line 197 (Fig 3b). There appears to be fewer cone-like nuclei compared with photoreceptor nuclei in Acute *Nrl* KO by Montana *et. al.* (PNAS 110:1732-7. 2013) despite earlier disruption of *Nrl* in this study. Is this because *Crxp-Nrl* transgenic mice were used?

We do not rule out the possibility that the copy number of *Nrl* alleles in *Crxp-Nrl* mice could have exceeded the editing capacity of CRISPR-*Nrl*, resulting in incomplete *Nrl* ablation in many transduced rods. However, it is also likely that acquisition of the cone-like morphology was affected by the timing of *NRL* abolishment and the epigenetic state of the *Nrl*-ablated rods, which varied hugely among the CRISPR-*Nrl* transduced cells. Therefore, not many *Nrl*-ablated rods displayed the typical cone-like morphology. It is hard to conclude that there were fewer cone-like nuclei in our study than in the previous study (Montana *et al.*, 2013, Proc Natl Acad Sci U S A, 110:1732), as a more typical cone-like morphology of nuclei (a larger euchromatin but smaller heterochromatin) was shown in our result (Fig. 3b). Additionally, the cone-like alteration following acute *Nrl* KO in the previous study also happened in some (but not a majority of) photoreceptors (please see Fig. 3C and 5D in the PNAS paper).

Line 215 (Fig 4a). *Grk1* (rhodopsin kinase) is expressed in both rod and cone photoreceptors. Please address this in the figure or in the text.

We thank the reviewer's comment and apologize for this oversight. As *Grk1* is expressed in both rods and cones, we have now removed it from Fig 4a and provided additional information in Supplementary Fig. 13.

Figs 5-7. The authors showed optomotor data only in RHO-P347S mice (Fig 7e). Have they tested optokinetic response in *Nrl*-disrupted *Rho*^{-/-} and *Rh10* mice (Figs 5 and 6)?

No, we have not tested optokinetic response in *Nrl*-disrupted *Rho*^{-/-} and *Rh10* mice. The main reason was the cataract formation in some *Rho*^{-/-} and *Rh10* mice due to ocular injection at P14, which would severely affect the optomotor response. However, ERG response, especially with high stimulus intensity, was not significantly affected by P14 injection.

Reviewer #3 (Remarks to the Author):

The authors develop a CRISPR Cas9 gene editing approach for retinal degeneration diseases. Specifically they target *Nrl* and show a rescue effect in 3 models.

We appreciate the reviewer's comments.

- Concerns exist with regard to how long Cas9 remains using AAV. The authors should check for long term Cas9 expression and potential for off target at later time points.

We understand the reviewer's concerns regarding the consequence of persistent expression of a bacterial protein with the ability of modifying genome in the retina. Although temporary Cas9 expression is desirable for our current approach, numerous pre-clinical and clinical studies have demonstrated that AAV vectors mediate long-term transgene expression in postmitotic cells. Our recent studies have shown that the AAV-mediated retinal expression of RPGR and RP2 proteins lasted for 18-24 months post vector administration in mice (Wu *et al*, 2015, Hum Mol Gen, 24:3956; Mookherjee *et al*, 2015, Hum Mol Gen, 24: 6446). We therefore believe that the AAV-delivered Cas9 was persistently expressed in the retina, and examination of its expression at a later time point in addition to the 6-week testing (Supplementary Fig. 1b) is not needed.

We followed the reviewer's suggestion and conducted an additional deep DNA sequencing to evaluate on-target and off-target events in flow-sorted tdTomato-positive photoreceptors collected at 9.5 months post CRISPR-*Nrl* treatment (Supplementary Fig. 6 and Supplementary Table 2). A similar indel rate (~93% vs. ~98%) was detected at the targeted *Nrl* locus. Again, no detectable off target events above background was noticed at the ten potential off-target loci. We added the following to Results.

“As AAV-delivered Cas9 and sgRNA-*Nrl* were likely persistently expressed, long-term on-target and off-target events were examined by deep sequencing at 9.5 months post treatment. The results revealed a slightly lower on-target mutation rate (~93%), and a similar pattern of sequence alterations with a predominant one adenosine insertion (~88%), compared with those at 3 months post treatment. Mutation rates at the predicted potential off-target sites were again not significantly higher than those in control groups, suggesting that long-term expression of CRISPR components may not necessarily impose a higher risk of off-targeting if the sgRNA is appropriately selected.”

Even though these results eased our concern to some extent on off-target genome modification, to ensure safety, we are not planning to directly apply the current version of AAV-CRISPR/Cas9 to human studies. The gene editing technology is rapidly evolving. In future human use, a controllable or temporarily expressed, more target-specific nuclease, such as high-fidelity Cas9 (Kleinstiver, *et al.* 2016, *Nature*, 529:490), should be employed.

-Is the gain in cone features maintained over time?

In the previous study (Montana *et al.*, 2013, *Proc Natl Acad Sci U S A*, 110:1732), up-regulation of cone-specific genes such as *Gnat2*, *Gnb3* and *Pde6c* was detected at 6 months following acute *Nrl* KO (>7 months of age), indicating that the cone-like features acquired in the *Nrl*-ablated rods can be maintained over time. Although a different approach was used in our study to achieve *Nrl* ablation, we believe that we would have obtained a similar result if examination of the expression profile of the CRISPR-*Nrl* transduced cells was conducted on older mice. CRISPR-mediated *Nrl* ablation (through cut and re-ligate by NHEJ) is irreversible. We expect the cells to remain *Nrl* null and maintain the cone-like phenotype for as long as the cell is viable. We therefore did not perform the experiment.

-Can the authors demonstrate that no other cell types are transduced? Difficult to see specificity of targeting in Fig 1g.

Fig. 1g was not intended to show cell-type specificity following AAV vector administration. We have now provided new results (Supplementary Fig. 1c) to show the cell type(s) transduced by the AAV8 vector under control of the rhodopsin kinase (RK) promoter. Previous studies (Allocca *et al.*, 2007, *J Virol*, 81:11372; Leberherz *et al.*, 2008, *J Gene Med*, 10:375) have shown that AAV8 vectors with an ubiquitous promoter (such as CMV) transduce RPE and photoreceptors efficiently, with occasional transduction of Muller and ganglion cells following subretinal injection in mice. Our results showed that under the control of the RK promoter, the AAV8-mediated tdTomato expression was limited to the photoreceptors (outer nuclear layer and inner segments). No expression was detected in other retinal layers. Specifically, no tdTomato expression was observed in the RPE65, SOX6 or BRN3A positive cells, indicating that the AAV8 vector with the RK promoter does not transduce RPE, Muller or ganglion cells.

We have modified the corresponding language in the text as below.

“The RK promoter-driven tdTomato expression was limited to photoreceptors and no expression was observed in retinal pigment epithelium (RPE) cells and cells in other retinal layers, confirming the photoreceptor-specificity of this AAV-delivered CRISPR/Cas9 system (Supplementary Fig.1c).”.

-The small changes seen in gene expression is less than predicted. This should be more clearly evaluated/discussed.

To address the reviewer’s concern, we performed a detailed re-analysis of transcriptome alteration following CRISPR-*Nrl* treatment (Supplementary Data 1, Supplementary Fig. 11 and 12), and compared the result with the transcriptome of P28 WT rods and S cone-like cells from *Nrl*^{-/-} mouse (Kim *et al*, 2016, Dev Cell, 37:520).

Only 146 genes exhibited differential expression (>2-fold & p-value ≤ 0.05) between control (CRISPR-EGFP treated) cells and CRISPR-*Nrl* treated cells (Supplementary Fig. 11a), strikingly lower than 6412 genes that were found differentially expressed between mature rods and S cone-like cells. We think that the low number of genes displaying differential expression following CRISPR-*Nrl* treatment could result from lack of plasticity of rods for reprogramming when ablation of *Nrl* took place.

We therefore were not surprised that only a few rod and cone specific genes showed statistically significant changes (>2-fold) after CRISPR-*Nrl* treatment. Results from the previous study (Montana *et al*, 2013, Proc Natl Acad Sci U S A, 110:1732) appeared to reveal expression changes in more genes than our current study. The discrepancy could be caused by different treatment for *Nrl* ablation (tamaoxifen-induced vs. AAV-CRISPR mediated) and different assay for expression (in situ hybridization vs. RNAseq). The variable results from in situ hybridization for a few genes in Montana *et al*’s study (compare Fig. S3 with Fig. S7 for *Pde6a*, Fig.3, Fig. S3, Fig. S6 and Fig. S7 for *Gnat2* in the PNAS paper) makes the comparison more difficult.

-The authors should include an epigenetic evaluation of changes in cell fate to further demonstrate acquisition of cone features.

We thank the reviewer’s suggestion. As we have discussed, the CRISPR-*Nrl* treatment only resulted in limited phenotype changes of the postmitotic rods, which we think have been adequately addressed by the results of transcriptome, protein analyses, functional and morphological evaluations (Fig. 3, 4). We speculate that the epigenetic changes of the CRISPR-*Nrl* treated cells would mainly involve a more active chromatin state allowing up-regulation of a small number of cone genes (such as *Gnb3* and *Arr3*). However, for a majority of rod and cone genes, their chromatin state may not be significantly affected. The previous study (Montana *et al*, 2013, Proc Natl Acad Sci U S A, 110:1732) has already shown that *Nrl* acute KO did not change the DNA methylation status of *Rho* and *Opn1sw* genes. An epigenetic evaluation requires a relatively large number of highly pure population of rods, which was not available in our study. It would help delineate how genes downstream of *Nrl* are regulated (or not regulated, as *Rho* and

Opn1sw genes in the Montana *et al* PNAS paper) by *Nrl* ablation, but may not provide significantly more meaningful information on cone feature acquisition than what we have shown in the manuscript.

-Is the heterogeneity seen due to global inefficiencies in targeting or clone differences upon targeting NRL? What is leading to the variability? Do you see lower or higher changes among different clones. If targeting is higher or lower in different clones does this correlate with changes in gene expression or do you see higher or lower. This would assist in directly correlating targeting to the cell fate and gene expression changes.

We thank the reviewer's excellent questions regarding the heterogeneity of CRISPR-*Nrl* treated cells (especially in the morphology of nuclei of CRISPR-*Nrl* treated retina). We do not think this was caused by the inefficiency in *Nrl* targeting, due to the following evidence.

1. Targeted DNA deep-sequencing revealed ~98% mutation rate in flow-sorted td-Tomato positive (sgRNA vector-transduced) photoreceptors following CRISPR-*Nrl* treatment (Fig. 2d), suggesting that both alleles of *Nrl* were ablated in a majority of photoreceptors transduced with both the Cas9 vector and the sgRNA-*Nrl* vector.
2. NRL were undetectable in a majority of photoreceptor nuclei after CRISPR-*Nrl* treatment (Fig. 2g).

Two major factors could have contributed to the variability of the outcome of the targeted photoreceptors.

1. Timing of *Nrl* ablation. As we discussed earlier, it may take a few days to a few weeks for the AAV vectors to express the Cas9 and the sgRNA, to find the target sequence and create DSB, and to repair the DSB. The DNA deep-sequencing and the NRL immunofluorescence analyses were conducted at ~3 months post treatment when this process was likely complete. However, this process could vary hugely among individual cells between P14 (administration time) and 3 months. If NRL abolishment happens earlier, the cell could be more likely to acquire cone features.
2. Plasticity of the postmitotic photoreceptor. This correlates to epigenetic state of the cell and could also vary among individual cells at a certain time point. In the study conducted by Joseph Corbo's group (Montana *et al*, 2013, Proc Natl Acad Sci U S A, 110:1732), *Nrl* ablation was achieved by tamoxifen-induced Cre expression, which could be faster and more uniform than the AAV-CRISPR approach. However, heterogeneity was still observed in the morphology of the converted rods (See Fig. 3C and 5D in the paper).

It would be very interesting if a correlation between *Nrl* ablation and cell fate/altered gene expression in individual photoreceptors can be established, but technically it might be hard to achieve. NRL staining can be conducted at an early time point (e.g. 3 weeks post treatment) when NRL-positive and NRL-negative cells co-exist in a retinal section, however, the varied epigenetic state of each cell would compound the results of cell fate and gene expression.

-Can you molecularly or immunocytochemically demonstrate changes in survival or cell death genes in NRL targeted cells?

We tested the active form of caspase-3, a commonly used marker for apoptosis which has been detected in degenerating photoreceptors in several mouse models of retinal degeneration. As shown in Supplementary Fig. 18, a few photoreceptors were stained positive in the representative retinal section of *RHO P347S* mouse receiving control vectors, while no positively stained photoreceptor was observed in the CRISPR-*Nrl* treated retina. Although a majority of photoreceptors in the control *RHO P347S* retina were not stained positive for active caspase-3 suggesting that caspase-3 independent cell death pathways could also be involved, our result demonstrate that ablation of *Nrl* inhibits the caspase-3 mediated photoreceptor apoptosis. We added the following to the text.

“The treatment appeared able to inhibit the caspase-3 mediated apoptosis pathway of the degenerating photoreceptors, although the caspase-3 independent pathways could also be involved in photoreceptor death in the *RHO-P347S* retina.”

- Is targeting one or both alleles efficiently or does this vary per cell?

We injected 2-fold higher amount of the Cas9 vector (5×10^9 vg/eye) than the sgRNA-*Nrl* vector (2.5×10^9 vg/eye), so that the sgRNA-expressing cells most probably expressed Cas9 as well. Targeted DNA deep-sequencing results revealed ~98% and ~93% indel rates in flow-sorted Td-Tomato positive (sgRNA vector-transduced) photoreceptors following CRISPR-*Nrl* treatment (Fig.2d and Supplementary Fig. 6). We therefore believe that both alleles of *Nrl* were ablated in a majority of photoreceptors co-transduced with the Cas9 and the sgRNA-*Nrl* vectors. This was elaborated by the undetectable NRL in a majority of photoreceptor nuclei after CRISPR-*Nrl* treatment (Fig. 2g). Accordingly we added the following to the main text.

“The deep sequencing (Fig. 2d,e, Supplementary Fig. 6) and the immunofluorescence analyses (Fig. 2g) collectively suggested that two *Nrl* alleles were disrupted in a majority of CRISPR-*Nrl* transduced cells.”.

We do not completely rule out the possibility that one allele of *Nrl* was not targeted, but this may only happen in a minority of cells.

REVIEWERS' COMMENTS:

Reviewer #1 (Remarks to the Author):

Authors have addressed all my comments with solid data; the manuscript has improved significantly. The approach of 'turning rods to cone-like' to rescue vision could be developed into a potential treatment for RP independent of genetic mutation. However, the long-term safety and efficacy of this AAV-CRISPR/Cas9 gene editing, especially intervention at the mid and late stage of degeneration need to be further investigated.

Reviewer #2 (Remarks to the Author):

The authors addressed all the issue properly.

Response to reviewers' comments

Reviewer #1 (Remarks to the Author):

Authors have addressed all my comments with solid data; the manuscript has improved significantly. The approach of 'turning rods to cone-like' to rescue vision could be developed into a potential treatment for RP independent of genetic mutation. However, the long-term safety and efficacy of this AAV-CRISPR/Cas9 gene editing, especially intervention at the mid and late stage of degeneration need to be further investigated.

We appreciate the comments and agree with the reviewer's point of view.

Reviewer #2 (Remarks to the Author):

The authors addressed all the issue properly.

We thank the reviewer's comment.